# Self-assembled hydrated copper coordination compounds as ionic conductors for room temperature solid-state batteries

Xiao Zhan[1], Miao Li[1], Xiaolin Zhao[2], Yaning Wang[2], Sha Li[1], Weiwei Wang[1], Jiande Lin [1], Zi-Ang Nan [1], Jiawei Yan [1], Zhefei Sun[1], Haodong Liu [3], Fei Wang [4], Jiayu Wan[5], Jianjun Liu [2] ✉, Qiaobao Zhang [1,6] ✉ & Li Zhang [1] ✉

As the core component of solid-state batteries, neither current inorganic solid-state electrolytes nor solid polymer electrolytes can simultaneously possess satisfactory ionic conductivity, electrode compatibility and processability. By incorporating efficient Li$^+$ diffusion channels found in inorganic solid-state electrolytes and polar functional groups present in solid polymer electrolytes, it is conceivable to design inorganic-organic hybrid solid-state electrolytes to achieve true fusion and synergy in performance. Herein, we demonstrate that traditional metal coordination compounds can serve as exceptional Li$^+$ ion conductors at room temperature through rational structural design. Specifically, we synthesize copper maleate hydrate nanoflakes via bottom-up self-assembly featuring highly-ordered 1D channels that are interconnected by Cu$^{2+}$/Cu$^+$ nodes and maleic acid ligands, alongside rich COO$^-$ groups and structural water within the channels. Benefiting from the combination of ion-hopping and coupling-dissociation mechanisms, Li$^+$ ions can preferably transport through these channels rapidly. Thus, the Li$^+$-implanted copper maleate hydrate solid-state electrolytes shows remarkable ionic conductivity (1.17 × 10$^{-4}$ S cm$^{-1}$ at room temperature), high Li$^+$ transference number (0.77), and a 4.7 V-wide operating window. More impressively, Li$^+$-implanted copper maleate hydrate solid-state electrolytes are demonstrated to have exceptional compatibility with both cathode and Li anode, enabling long-term stability of more than 800 cycles. This work brings new insight on exploring superior room-temperature ionic conductors based on metal coordination compounds.

Solid-state lithium-metal batteries (LMBs) comprising Li metal anode and non-combustible solid-state electrolytes (SSEs) are widely recognized as one of the most promising next-generation energy storage systems with both high energy density and superior safety[1–3]. As the core component, it is of critical importance to develop high-performance SSEs with high ionic conductivity, extraordinary chemical/electrochemical stability, exceptional electrode compatibility and ease of processing[4–8].

Generally, SSEs can be categorized into two groups: inorganic ceramic electrolytes and organic polymer electrolytes. Inorganic SSEs such as Li$_{10}$GeP$_2$S$_{12}$ (LGPS), Li$_7$La$_3$Zr$_2$O$_{12}$ (LLZO) and Li$_{1+x}$Al$_x$Ti$_{2-x}$(PO$_4$)$_3$ (LATP) typically exhibit high ionic conductivity of greater than

$10^{-3}$ S cm$^{-19-14}$. The internal Li$^+$ ion transport in these compounds predominantly occurs via collective diffusion mechanism along channels in their crystal lattice[4]. For instance, LGPS shows a one-dimensional (1D) channel along the *c*-axis, while LATP and LLZO have characteristic three-dimensional (3D) Li$^+$ transport channels[15]. Despite their high ionic conductivity, these inorganic SSEs still face daunting challenges such as low oxidation stability, poor processability, water sensitivity, inferior electrode compatibility and etc[16,17]. In contrast, solid polymer electrolytes (SPEs) based on polymers such as poly(ethylene oxide) (PEO), polyacrylonitrile (PAN), and poly(vinylidene fluoride) (PVDF) offer multiple advantages, including good flexibility, ease of processing, and strong affinity with the electrode interface[18-24]. The conduction of Li$^+$ ions in SPEs occurs through a coupling-dissociation process between the Li$^+$ ions and the polar groups present in the polymer chain segments, such as -O- in PEO, -CN in PAN, and -F in PVDF[4,25]. However, these chain segments in high-crystallinity polymers are typically entangled with each other, resulting in limited space for free Li$^+$ movement and poor ionic conductivity ($10^{-7}$-$10^{-5}$ S cm$^{-1}$), as well as low Li$^+$ transference numbers at room temperature[6,26]. While composite solid-state electrolytes that combine the benefits of inorganic SSEs and organic SPEs have demonstrated certain improvements over individual electrolytes, the two types of SSEs are not perfectly integrated. This can result in an incompatible two-phase interface, which becomes a bottleneck that limits the transport of Li$^+$ ions[27-30]. Therefore, it is highly desirable but remains challenging to develop a single-component SSE that can synergistically leverage the advantages of inorganic and organic SSEs.

In 2021, Hu et al. pioneered a distinctive type of organic-inorganic hybrid SSEs based on the Cu$^{2+}$-coordinated cellulose polymer[31]. Rapid Li$^+$ conduction is facilitated through an ion-hopping mechanism that utilizes polar groups and bound H$_2$O molecules in 1D channels, which are expanded and activated by Cu$^{2+}$ coordination. This structural characteristic could be utilized to design materials with ordered channel structures and rich polar functional groups, such as metal-organic framework (MOF) compounds. However, studies have shown that intrinsic MOFs with ultra-high porosity are not satisfactory Li$^+$ conductors due to the large spacing between Li$^+$ host nodes in MOFs, which makes it difficult to form a feasible 3D percolation network[32-35].

Herein, we demonstrate that another important class of previously unexplored organic-inorganic hybrid materials, metal coordination compounds, can serve as exceptional room-temperature Li$^+$ ion conductors by screening ligands with appropriate size and components. Specifically, the unique two-dimensional (2D) copper maleate hydrate (CuMH) nanoflakes are synthesized through a batch-produced self-assembly process using maleic acid (MA) as the precursor ligand (Fig. 1a). The CuMH nanoflakes are endowed with highly-ordered 1D channels interconnected by Cu$^{2+}$/ Cu$^+$ nodes and MA molecules, as well as rich COO$^-$ groups and structural water within the channels (Fig. 1e). Benefitting from the appropriate distance between oxygen-containing sites, Li$^+$ ions can preferably diffuse along a continuous percolation network where ion-hopping and coupling-dissociation modes are organically integrated. As such, the Li$^+$-implanted CuMH (Li-CuMH) SSE (Fig. 1d) shows multiple remarkable features, including prominent ionic conductivity ($1.17 \times 10^{-4}$ S cm$^{-1}$), low activation energy (0.123 eV), high Li$^+$ transference number (0.77), and a wide electrochemical window of 0-4.7 V at room temperature. More impressively, the as-fabricated Li-CuMH SSE can guarantee long-life Li-plating/stripping for over 2000 h with an areal capacity of 0.3 mAh cm$^{-2}$ at 0.3 mA cm$^{-2}$. And the solid-state LiFePO$_4$/Li-CuMH SSE/Li full cell can maintain over 80% of its capacity at 0.5 C (1 C = 170 mAg$^{-1}$) for long-term cycling life of 574 cycles at 25 °C while maintaining an impressive rate performance, meeting practical performance standards. Furthermore, the soft-package LiFePO$_4$/Li full cell utilized Li-CuMH SSE also demonstrates desirable cycling stability and can work in harsh conditions such as bending and cutting, holding potentiality for practical applications.

Additionally, the mechanistic origins of the enhanced performance of Li-CuMH SSE in solid-state LMBs are explicitly revealed through electrochemical measurements, substantial ex situ microscopic and spectroscopic techniques coupled with theoretical calculations. Our concept of constructing metal coordination compounds with high ionic conductivity through rational screening of metal ions and ligands will open a broad space for the design of high-performance SSEs at room temperature.

## Results and Discussion

### Synthesis of CuMH nanoflakes

The synthetic procedure of CuMH nanoflakes is schematically shown in Fig. 1a (see the experimental section for details). Briefly, a fresh copper (Cu) foil was directly immersed in an acetonitrile (AN) solution containing an appropriate amount of maleic acid (MA) and lithium nitrate (LiNO$_3$) under atmospheric environment, in which MA not only acts as ligand, but also releases H$^+$ to cooperate with NO$_3^-$ to promote the dissolution of copper (the significant acceleration effect of NO$_3^-$ on CuMH production is shown in Fig. S1). Then, Cu atoms in situ coordinated with MA ligands and H$_2$O molecules and assembled into blue precipitates in a bottom-up manner. Notably, the dissolution of Cu and the self-assembly process of CuMH crystals are spontaneous, highly-efficient, and easy to be scalably produced (as demonstrated in the digital image of 50 g CuMH sample in Fig. 1a). Figure 1b discloses that CuMH exhibits a regular disc-shaped structure with an average diameter of 2-3 μm, and the cross-sectional SEM image further reveals that these micro-disks are tightly stacked by layered CuMH nanosheets (Fig. 1c). As a reference, CuMH prepared through traditional hydrothermal methods using Cu(NO$_3$)$_2$ as the precursor only exhibits irregular nanosheet structure (Fig. S2).

To study the valence information of Cu cations in the CuMH crystal, XPS and Auger Cu LMM spectra of the CuMH sample were collected and analyzed (Fig. S3). The binding energy (BE) peak at ~935 eV is assigned to Cu$^{2+}$ species, accompanied by the characteristic Cu$^{2+}$ shakeup satellite peaks (938-945 eV)[36]. The BE peak at ~933 eV clearly demonstrates the existence of Cu$^+$ or Cu$^0$ species (Fig. S3a)[37,38]. And the Auger Cu LMM spectrum further confirms the presence of Cu$^+$ at ~570 eV (Fig. S3b)[39]. In contrast, the CuMH film prepared from Cu(NO$_3$)$_2$ only shows the characteristic peak of Cu$^{2+}$ species (Fig. S4). These results manifest that starting from Cu$^0$ foil is a prerequisite for obtaining CuMH film containing Cu$^+$ cations. Furthermore, the ratio of Cu$^+$ and Cu$^{2+}$ content in our CuMH is calculated by integrating the area of the corresponding peaks in Fig. S3b, which shows that the proportion of Cu$^+$ and Cu$^{2+}$ in CuMH is 58.8% and 41.2%, respectively. The appearance of Cu$^+$ ions indicates that some undissociated H$^+$ ions exist on the carboxylic acid that coordinates with Cu$^+$ ions, resulting in the presence of (HOOCC$_2$H$_2$COO)$^-$ in the structure of the CuMH crystal, which can be confirmed by liquid $^1$H nuclear magnetic resonance ($^1$H NMR) measurements (Fig. S5). And this coordination structure is also found in other Cu-MOF materials containing carboxylic acid[40,41]. Powder X-ray diffraction (PXRD) pattern of the as-obtained CuMH powder is shown in Fig. S6 and single-crystal XRD pattern of the CuMH grown in water perfectly matches with the standard powder diffraction card (PDF#49-2453). To further determine the crystal structures of Cu$^{I/II}$MH and Li-Cu$^{I/II}$MH samples, the General Structure Analysis System (GSAS) program was used to solve and refine the corresponding PXRD patterns by the Rietveld method[42], which confirms the existence of protons in the CuMH crystal and the Li$^+$/H$^+$ exchange reaction for the Li-CuMH sample (Fig. S7 and Table S1). Moreover, the crystal structures of CuMH and Li-CuMH were further calculated and presented in Fig. S8 (Supplementary Data 1). The Cu (II) ion possesses a square pyramidal coordination environment, and the apical oxygen atoms originate from the localized structural water molecules. The base atoms come from two oxygen atoms of a single maleate chelating ligand, and two oxygen atoms come from two additional maleate groups. The Cu (I)

ion has a similar square pyramidal coordination environment as the Cu (II) ion, with the difference that one of the coordination oxygens is a hydroxyl oxygen, not a conjugated carbonyl oxygen. Each maleate group is bonded to three Cu atoms and forms a polymeric monolayer. These polymeric monolayers are closely connected by strong hydrogen bonds between water molecules and the carbonyl oxygen atoms to form a periodic stacked structure (Fig. S9)[43]. This explains the experimental finding that the CuMH micro-disks are composed of tightly packed nanosheets from the perspective of crystal structure (Figs. 1b, c).

## Fabrication and characterization of Li-CuMH SSEs

Afterwards, a flexible and compact CuMH film with a thickness of *ca*. 30 μm was fabricated by rolling a homogeneous slurry of CuMH powders and polytetrafluoroethylene (PTFE) emulsion (Fig. S10). Subsequently, the Li-CuMH membrane was ultimately accomplished by fully impregnating in a non-aqueous electrolyte containing lithium bistrifluoromethanesulfonimide (LiTFSI) salt to achieve Li[+] ion implantation (Fig. S11). As demonstrated in Figs. S12, the Li[+] ion undergoes an ion-exchange reaction with the H[+] ions on the carboxylic acid to form charge-equilateral Li-CuMH model during the Li[+] impregnation process. The as-obtained Li-CuMH SSE films were then vacuum dried for at least 12 h to thoroughly remove the residue solvent and moisture prior to use. Figure 1d presents a digital photograph of a typical 6 cm × 6 cm-sized Li-CuMH SSE film, revealing its exceptional processability.

More importantly, a 1D channel rich in carboxyl groups and structural water appears between the stacked skeletons, providing a great possibility for the rapid Li[+] ion transport (Fig. 1e).

Fourier transform infrared (FT-IR) measurements were performed to analyze the chemical environment in the CuMH and Li-CuMH crystals, as well as the existing form of structural water molecules. As illustrated in Fig. S13, the FT-IR peaks located at 1540, 1580/1433, 682 /620 cm[−1] can be identified to -COOH group, COO[−] group and Cu-O bond of the CuMH sample, respectively[44,45]. Note that the characteristic peak of -COOH group completely disappears in the FT-IR spectrum of the Li-CuMH sample and the new peak at 870 cm[−1] is assigned to the Li-O bond[46]. The broad band in the wavenumber range of 3200 to 3500 cm[−1] can be identified as the O-H stretching vibration of structural water, and the band at around 1654 cm[−1] corresponds to the H-O-H bending mode in the lattice (Fig. S13). The FT-IR spectra confirmed the coexistence of -COOH, COO[−] groups and structural $H_2O$ in the CuMH lattice, which is highly consistent with what we found in [1]H NMR analysis (Fig. S5).

After the sufficient Li[+] ion implantation, the structural evolution of the Li-CuMH relative to the pristine CuMH was evaluated by small-angle X-ray scattering (SAXS) measurements. As demonstrated in Figs. 2a, b and S14, the CuMH sample exhibits two strongest peaks at the angstrom scale of $d_1 \approx 7.20$ Å and $d_2 \approx 4.26$ Å, while the Li-CuMH sample presents two strongest peaks at the angstrom scale of $d_1 \approx 7.25$ Å and $d_2 \approx 4.30$ Å. Apparently, after soaking in the organic liquid electrolyte, the

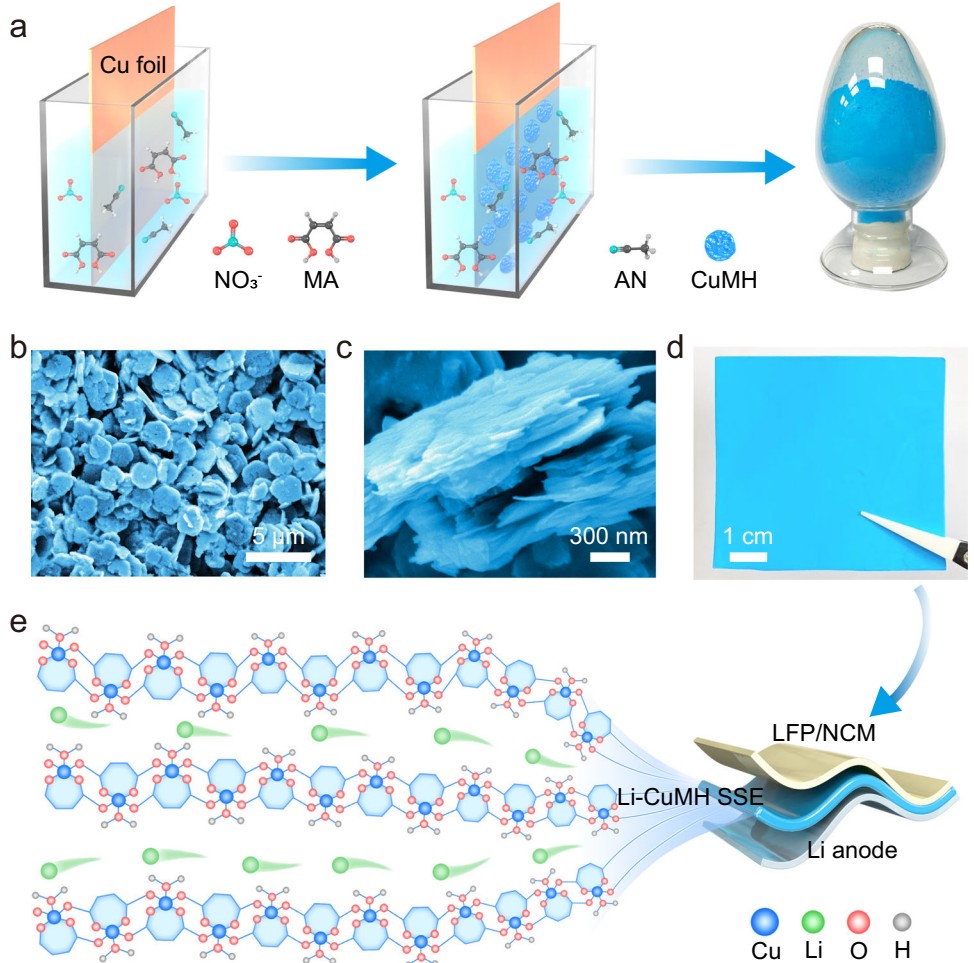

**Fig. 1 | Synthesis and schematic diagram of the Li+ ion transport channels in Li-CuMH. a** Schematic of the self-assembly synthesis route of CuMH powders. NO$_3^−$, MA, AN, CuMH are represented as nitrate ion, maleic acid, acetonitrile, and copper maleate hydrate, respectively. **b** Top view and **c** side view SEM images of 2D CuMH nanoflakes. **d** Digital photograph of a typical 6 cm × 6 cm-sized Li-CuMH SSE film. **e** Schematic diagram of 1D Li[+] ion transport pathways in Li-CuMH.

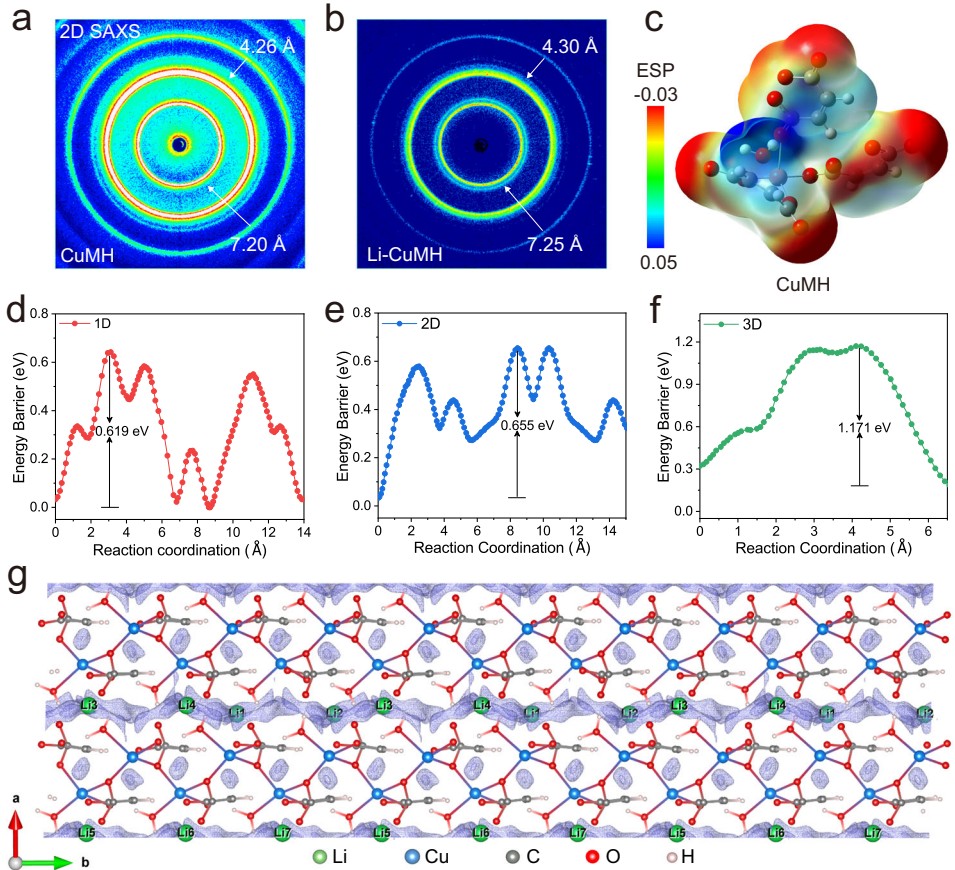

**Fig. 2 | Structure and Li+ ion migration pathways.** 2D SAXS patterns of **a** CuMH and **b** Li-CuMH samples. **c** Electrostatic potential (ESP) distribution of the CuMH unit. Energy profiles of **d** 1D, **e** 2D and **f** 3D possible Li+ ion migration pathways in Li-CuMH. **g** The most favorable Li+ migration pathway along the [010] direction of the b-axis, viewed as the light purple isosurface of constant $E_{BVSE(Li)}$ superimposed on the crystal structure.

angstrom-scale channels of CuMH accordingly expand and the corresponding peak intensity weakens, convincingly confirming the effective embedding of Li+ ions into CuMH crystals. Fig. S15 further shows the X-ray diffraction (XRD) pattern of CuMH and Li-CuMH SSE films and the corresponding enlarged spectra in the 2-theta range of 11.5° and 13°. The characteristic peak corresponding to the (001) crystal plane changed significantly after the sufficient Li+ ion implantation, mainly manifested as the peak position shifting negatively from 12.27° to 12.21°, accompanied by a decrease in peak intensity. This clearly indicates that the crystal spacing indeed expands with the implantation of Li+ ions, and the diffraction intensity decreases accordingly. Moreover, the crystal structures of Li-CuMH SSE films after soaking in the non-aqueous electrolyte containing LiTFSI salts for different times were further investigated (Fig. S16). Apparently, the characteristic peaks located at around 12.3° continue to shift negatively, accompanied by a continuous decrease in peak intensity, indicating that Li+ ions are indeed constantly intercalated into crystal interlayers. This is highly consistent with what we found in Fig. S15. Electrochemical impedance spectroscopy (EIS) measurements were further performed to estimate the change in ionic conductivities of Li-CuMH SSE films after soaking for various durations. As Fig. S17 demonstrates, the Li-CuMH SSE films show significantly reduced polarization as the soaking time increased, and the ionic conductivity increases from $2.5 \times 10^{-5}$ to $1.1 \times 10^{-4}$ S cm$^{-1}$ when the soaking time increase from 0.5 to 24 h. Given the above, the correlation between the crystal structures and ionic conductivities of Li-CuMH SSEs with different Li+ implantation times convincingly indicates that Li+ ions can intercalate into the bulk crystal

structure, ultimately resulting in varying degrees of ionic conductivities.

## Li+ conduction in the Li-CuMH SSE

The structural characterization of Li-CuMH revealed a well-defined layered structure, with 1D channels that are rich in polar carboxyl groups and confined structural $H_2O$ (Fig. S18). These characteristics provide abundant Li+ ion hopping sites and continuous pathways, making Li-CuMH a promising candidate for an innovative SSE with significant properties. Density functional theory (DFT) calculations were carried out to reveal the interaction mechanism between Li+ ions and COO$^-$ and $H_2O$. As manifested in Fig. 2c and S19, the electrostatic potential (ESP) mappings indicate the high local electronegativity of O atoms from the maleate unit and structural $H_2O$ in the CuMH crystal. These sites act as Lewis bases and can preferentially bind to Li+ ions, as demonstrated by the calculations[47]. The bond valence site energy (BVSE) method was used to calculate the possible migration pathways of Li+ ions in Li-CuMH and their corresponding migration energy barriers, as shown in Fig. S20−22. Figure 2d, f illustrate the migration barriers of Li+ ions along the labeled 1D, 2D, and 3D paths. Although the overall migration energy barrier is higher than the experimental value, the BVSE method provides an efficient screening tool for assessing the relative height of the barriers[48]. The direction of the [Li1-Li2-Li3-Li4] chain along the b-axis was found to be the most favorable 1D migration path for Li+ ions (Fig. S20), with a minimum migration energy barrier of 0.619 eV. Li+ ions were observed to migrate along the [Li1-Li2-Li3-Li4] chain and interconnect with the [Li1-Li1] chain to form a 2D migration path in the bc plane (Fig. S21), with a corresponding migration energy barrier of 0.655 eV. In contrast, the 2D path connects the remaining

active sites into a 3D path (Fig. S22), but the migration energy barrier is relatively high at 1.171 eV. The BVSE method was also utilized to provide insight into the possible Li$^+$ ion diffusion pathway in the dimensionality of diffusion, as demonstrated by the isosurface of Li$^+$ ion densities (view as the light purple isosurface of constant $E_{BVSE\,(Li)}$ for Li in the model, superimposed on the crystal structure). Figure 2g explicitly shows the 1D Li$^+$ ion migration pathway along the [010] direction of the *b*-axis in the bond valence energy map. Interestingly, the actual trajectory of Li$^+$ ions exhibits periodic fluctuations biased towards oxygen atoms due to the local electronegativity of COO$^-$ and H$_2$O and their partial binding with Li$^+$ ions. This result is consistent with the high affinity of COO$^-$ and H$_2$O with Li$^+$ ions as indicated by DFT calculations.

To further investigate the Li$^+$ diffusivity and activation energy of Li-CuMH SSE, ab initio molecular dynamics (AIMD) simulations were conducted. Mean square displacement (MSD) plots from 350 to 750 K are shown in Figs. S23a-e, including Li, H, O, C and Cu (Supplementary Data 1). The MSD plots indicate that Li$^+$ ions exhibit the fastest migration coefficient of $1.53 \times 10^{-5}$ cm$^2$ s$^{-1}$, while H$_2$O, COO$^-$, HC = CH, and Cu (II) in the Li-CuMH backbone move much more slowly. Moreover, the activation energy ($E_a$) of the Li-CuMH is calculated to be 0.43 eV (Fig. S23f). Both $D_{Li^+}$ and activation energy support the fast

kinetic and low-barrier Li$^+$-ion migration pathways in the Li-CuMH SSE, which is attributed to the well-defined channels interconnected by Cu (II) and COO$^-$, with the participation of structural water molecules.

The presence of structural water in Li-CuMH (theoretical content 9.2 wt.%) is critical for determining the overall ionic conductivity and the working voltage window. To evaluate the effect of structural water on the ionic conductivity and electrochemical stability, Li-CuMH tablets with different structural water contents were obtained by soaking sintered CuMH powders into the non-aqueous electrolyte containing LiTFSI salts to accomplish Li$^+$ ion exchange and then pressed into thick tablets through adopting a cold pressing method (Fig. 3a, b). As shown in Fig. 3c, the ionic conductivity of the Li-CuMH SSEs decreases with the loss of structural water, which indicates the essential contribution of structural water to the Li$^+$ ion transport in Li-CuMH SSEs. Note that the crystal structure of Li-CuMH deteriorates when all the structural water is lost, fully indicating that structural water is an indispensable component in crystal structure of Li-CuMH (Fig. S24).

Furthermore, the Li-CuMH SSE exhibits high stability against Li metal, indicating that the structural water in the coordination state does not possess the chemical activity of free liquid water (Fig. S25). Radial distribution function (RDF) analysis was performed to analyze

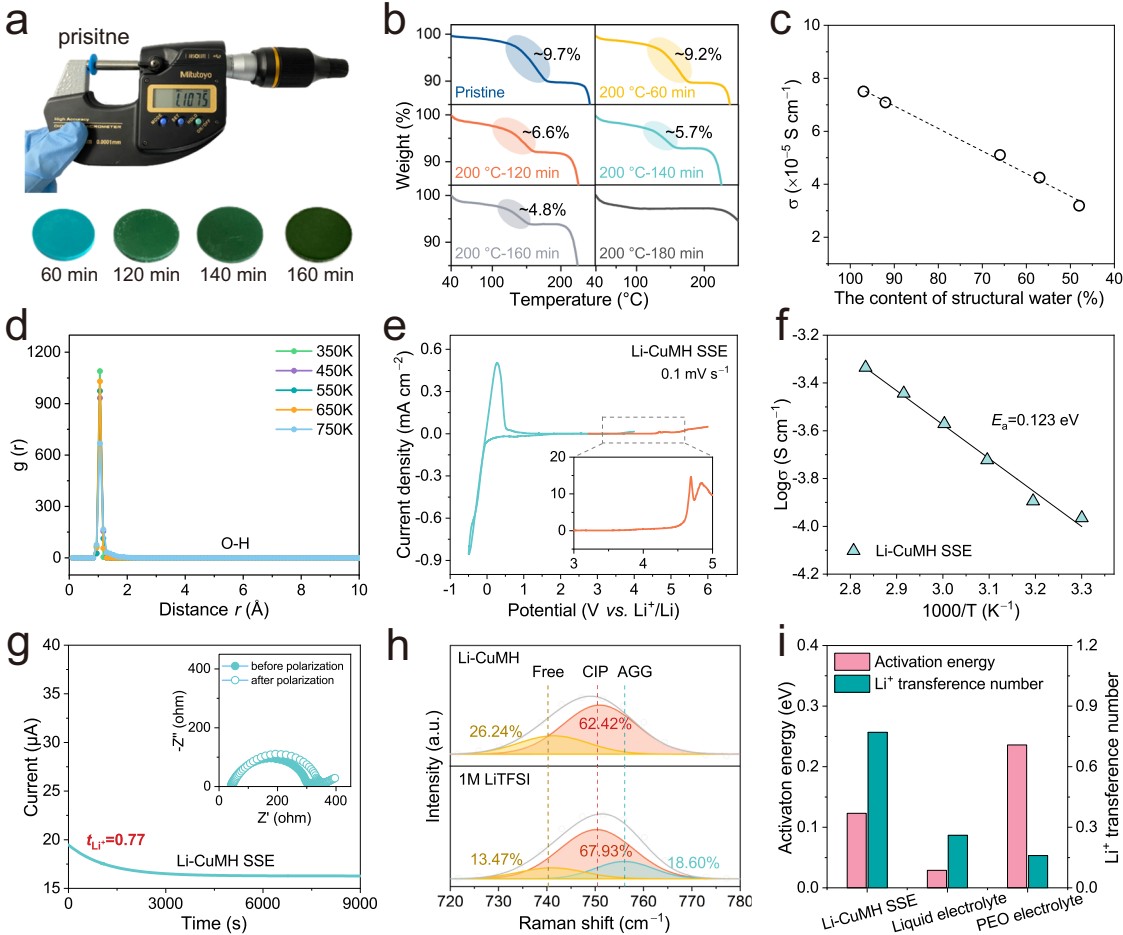

**Fig. 3 | Li$^+$ conduction in the Li-CuMH SSE. a** Digital photographs of Li-CuMH tablets formed by pressing CuMH powder sintered at 200 °C for different times and fully implanted with Li$^+$ ions. The thickness of Li-CuMH tablets was controlled around 1.1 mm. **b** TGA curves of CuMH powders after sintering at 200 °C for different times. **c** Ionic conductivities of Li-CuMH tablets formed by pressing CuMH powder sintered at 200 °C for 60, 120, 140, 160 and 180 min and fully implanted with Li$^+$ ions. **d** Radial distribution function of O-H in the structural H$_2$O using the trajectory obtained by MD calculation. **e** Cyclic voltammetry profile of the Li-CuMH

SSE between −0.5 and 4 V, and linear sweep voltammetry curve of the Li-CuMH SSE in a potential range of open circuit potential to 6 V (the inset is the corresponding enlarged image). **f** Arrhenius plot of the Li-CuMH SSE. **g** Li$^+$ transference number measurement of the Li-CuMH SSE. The inset is the Nyquist plot of Li/Li-CuMH SSE/ Li symmetric cell before and after polarization. **h** Raman spectra of the TFSI$^-$ vibration of 1 M LiTFSI in liquid DOL/DME electrolyte and Li-CuMH SSE. **i** Comparison of activation energy and Li$^+$ transference number of Li-CuMH SSE, organic liquid electrolyte and solid polymer PEO electrolyte.

the radial distribution of hydrogen bonds (Fig. 3d), where the O-H bonds of 1.05 Å in $H_2O$ from the Li-CuMH structure show the highest aggregation at different temperatures and the values of hydrogen bonds are approximately the same, convincingly indicating that the structural $H_2O$ is highly stable in the framework and does not offer proton conductivity. To further investigate the electrochemical stability and operating voltage window of Li-CuMH SSEs, cyclic voltammetry (CV) and linear sweep voltammetry (LSV) measurements were carried out. The results in Fig. 3e reveal a wide electrochemical stability window of 0–4.7 V for Li-CuMH SSE, and the reduction and oxidation peaks observed around 0 V correspond to the Li plating/stripping process. It is worth noting that the maximum acceptable current density of 15 $\mu A\,cm^{-2}$ (inset in Fig. 3e) defines the upper limit of the 4.7 V voltage[49]. These findings demonstrate the high electrochemical stability and a wide operating potential window of Li-CuMH SSEs, which can pave the way for their potential use in high-performance energy storage devices[49]. In order to gain an in-depth understanding of the high-voltage resistance of the Li-CuMH, the oxidation stability was analyzed by calculating highest occupied molecular orbital (HOMO) value through DFT simulations[50]. As manifested in Fig. S26, the HOMO energies of CuMH and Li-CuMH are −7.63 and −7.02 eV, respectively. The high antioxidation ability of the Li-CuMH is mainly attributed to the strong electron-withdrawing effect of the oxygen-containing groups. These results indicate that the Li-CuMH SSEs possess excellent compatibility with Li metal and most high-voltage cathode materials, owing to their high electrochemical stability and wide working potential window. This is highly consistent with the experiments shown in Fig. 3e and Fig. S25.

Electrochemical impedance spectroscopy (EIS) of the Li-CuMH SSE was recorded in the temperature range of 25 to 80 °C (Fig. S27a), and the corresponding Arrhenius plots reflect the temperature-dependent ionic conductivity (Fig. 3f). The room-temperature ionic conductivity of the Li-CuMH SSE is $1.17 \times 10^{-4}\,S\,cm^{-1}$ and the calculated activation energy ($E_a$) is 0.123 eV. The Li$^+$ transference number ($t_{Li^+}$) of the Li-CuMH SSE was estimated by chronoamperometry and EIS techniques. It is well recognized that low $t_{Li^+}$ is easily to produce ion concentration gradients in the electrolyte bulk and at the electrode/electrolyte interface, which may block the ionic transport path and lead to uneven Li deposition on the Li metal surface. Impressively, our Li-CuMH SSE shows a high $t_{Li^+}$ value of 0.77, revealing a rapid Li$^+$ ion transport (Fig. 3g). In order to identify specific forms of TFSI$^-$ coordination to the lithium cations, the expansion and contraction modes of the entire TFSI$^-$ anion at 750 $cm^{-1}$ was further analyzed using Raman spectroscopy (Fig. 3h), which could produce large polarization changes[51]. This wave band can be divided into three vibrational components located at 740, 750 and 756 $cm^{-1}$, respectively derived from free anion, TFSI$^-$ coordination to one Li$^+$ cation (CIP) and TFSI$^-$ coordination to two or more Li$^+$ cations (AGGs)[52]. Apparently, more free TFSI$^-$ anions can be released from Li-CuMH compared with the organic liquid electrolyte containing 1 M LiTFSI, indicating an increased dissociation content of LiTFSI salt in Li-CuMH. Interestingly, the AGGs signal completely disappears in the Li-CuMH SSE spectrum, further demonstrating the optimized dissociation effect. FT-IR analysis could also confirm the increase in free TFSI$^-$ content in the Li-CuMH SSE (Fig. S28). These results provide another supporting evidence that electronegative COO$^-$ groups and structural water in Li-CuMH can preferentially coordinate with Li$^+$ ions, promoting the dissociation of LiTFSI salts.

For comparison, the activation energy and Li$^+$ transference number of traditional organic liquid electrolyte and solid polymer PEO electrolyte were evaluated and compared at room temperature (Fig. S29 and 30). The preparation of PEO electrolyte and the cell assembly using liquid electrolyte and PEO are illustrated in experimental section (see the experiment section for details). As shown in Fig. 3i and Table S2, although the $E_a$ of the liquid electrolyte is as low as 0.029 eV, its $t_{Li^+}$ only reaches 0.26 owing to the strong solvation effect. By contrast, the PEO electrolyte shows high $E_a$ (0.236 eV) and low $t_{Li^+}$ (0.16) at the same time, reflecting its very poor Li$^+$ ion conductivity at room temperature.

## Durable Li-metal electrodes enabled by Li-CuMH SSE

Li/Li symmetric cells with Li-CuMH SSEs (denoted as Li/Li-CuMH SSE/Li) were assembled to evaluate the compatibility of the Li-CuMH SSE with Li metal. Critical current density (CCD) of the Li-CuMH SSE was determined by galvanostatic cycling with step-increased current densities (Fig. 4a). 0.1 $mA\,cm^{-2}$ was set as the step value, and the holding time was an hour per step. Apparently, the voltage of the Li/Li-CuMH SSE/Li cell suddenly drops at 1.0 $mA\,cm^{-2}$, suggesting the failure of the Li/Li symmetric cells. The CCD value of the Li/Li-CuMH SSE/Li cell is therefore determined to be 0.9 $mA\,cm^{-2}$, and the corresponding low overpotential is also significantly superior to most reported solid-sate electrolytes (Fig. 4b and Table S3)[31,53–58]. The long-term Li plating/stripping reversibility was further investigated at various current densities and areal capacities. At the current density of 0.3 $mA\,cm^{-2}$ and areal capacity of 0.3 $mAh\,cm^{-2}$, the Li/Li-CuMH SSE/Li symmetric cell shows an ultra-long durability over 2000 h at room temperature (Fig. 4c). During the whole Li stripping/plating process, the cell overpotential maintains stable, basically less than 100 mV. Even at a higher current density of 0.5 $mA\,cm^{-2}$, the symmetrical cell also maintains a high cycling stability of up to 600 h (Fig. 4d). Note that the gradual overpotential increase in the first 20 cycles is attributed to the activation process of interface between the Li-CuMH SSE and Li metal electrode until a stable interface is established. For comparison, Li/Li symmetric cells using the conventional liquid electrolyte and PEO electrolyte began to fail after only 350 h (Fig. S31a) and 400 h (Fig. S31c) at 0.2 $mA\,cm^{-2}$, respectively. The above results convincingly verify that Li-CuMH SSE has extraordinary interfacial compatibility and electrochemical stability compared to liquid electrolyte and solid polymer electrolyte.

To gain insight into the chemical/electrochemical stability of the Li-CuMH SSE/Li interface, Nyquist plots of Li/Li-CuMH SSE/Li cells after various cycle numbers at 0.5 $mA\,cm^{-2}$ were summarized in Fig. 4e. Fig. S32 demonstrates the equivalent circuit and module description of a typical Nyquist plot, in which $R_b$ represents the bulk resistance (high frequency), $R_{SEI}$ is the SEI resistance (medium frequency), $R_{CT}$ stands for charge-transfer resistance (low frequency) and $W$ is the Warburg impedance (linear line). The interface resistance between the Li-CuMH SSE and Li metal ($R_{SSE/Li}$) is the sum of $R_{SEI}$ and $R_{CT}$[59]. The fitted results of all Nyquist plots are listed in Table S4. Apparently, $R_b$ is almost constant throughout the cycling process, while $R_{SEI}$, $R_{CT}$ and $R_{SSE/Li}$ evolve accordingly with the cycle. In particular, $R_{SEI}$ and $R_{SSE/Li}$ show a similar trend, slowly increasing in the initial 20 cycles, and then gradually decreasing, which is highly consistent with the overpotential change of the Li/Li-CuMH SSE/Li cell at 0.5 $mA\,cm^{-2}$ (Fig. 4d).

$^7$Li solid-state magic angle spinning nuclear magnetic resonance (ssMAS NMR) spectra were collected to study the local Li$^+$ ion environment in the Li-CuMH before and after cycling (Fig. S33). For the cycled Li-CuMH SSE, the $^7$Li full-width at half-maximum (FHWM) decreases compared with the pristine Li-CuMH SSE, indicating an enhanced Li$^+$ ion mobility[60]. Moreover, the $^7$Li resonance of the pristine Li-CuMH is 1.60 ppm and shifts upfield after cycling, representing a change in the polarization environment of the Li$^+$ ions in the Li-CuMH. This change usually indicates a strong interaction between Li$^+$ ions and polar groups in the Li-CuMH SSE, which enhances the ionic conductivity[61,62]. Flammability tests were further performed to confirm the safety of the Li-CuMH SSE. As shown in Fig. 4f, the Li-CuMH SSE is absolutely flame retardant, while the glass fiber carrying the liquid electrolyte is easy to ignite. Undoubtedly, the flame-retardant

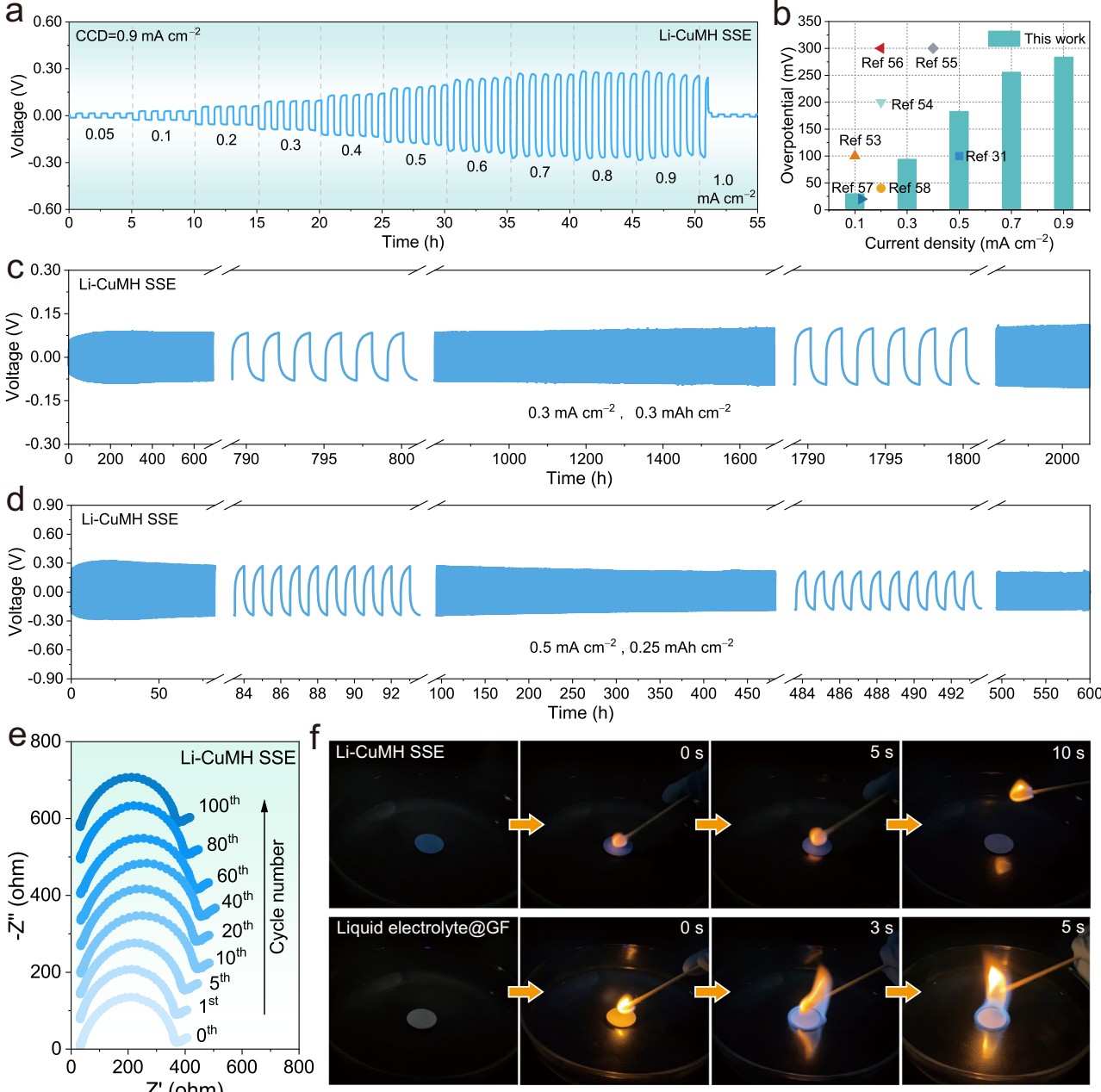

**Fig. 4 | Durable Li-metal electrodes enabled by Li-CuMH SSEs. a** Critical current density (CCD) measurement of the Li/Li-CuMH SSE/Li symmetric cell. **b** Comparison of CCD and overpotential values between the Li-CuMH SSE and reported solid-state electrolytes[31,53–58]. Long-term Li plating/stripping reversibility of Li/Li-CuMH SSE/Li symmetric cells at current densities of **c** 0.3 mA cm⁻² and **d** 0.5 mA cm⁻². **e** Ex situ Nyquist plots of Li/Li-CuMH SSE/Li cells after various cycle numbers at 0.5 mA cm⁻². **f** Flammability tests of the Li-CuMH SSE and glass fiber separator carrying liquid electrolyte.

properties of SSEs based on hydrated coordination compounds will greatly expand their application potential.

## Li-CuMH SSE-based solid-state full batteries

To evaluate the electrochemical performance of Li-CuMH SSEs in LMBs, full batteries were assembled with a LiFePO₄ (LFP) cathode and a Li metal anode (denoted as LFP/Li-CuMH SSE/Li). Notably, Li-CuMH powders were also added into the cathode laminates to guarantee an efficient ion-conducting network. Briefly, Li-CuMH powders were dispersed into the N-Methylpyrrolidone (NMP) solution to obtain a homogeneous emulsion (inset in Fig. 5a). The LFP cathode laminates were fabricated by casting the slurry of LFP, super P carbon, PVDF and Li-CuMH (Fig. 5a) onto the Al foil in a certain proportion. Note that the

active material mass loading was around 4 mg cm⁻². Moreover, the Li-CuMH SSE was pressed on the LFP laminate to ensure close interfacial contact. As displayed in Fig. 5b, the Li-CuMH SSE shows a thickness of *ca.* 30 μm and a tight contact with the LFP cathode without any visible chinks at the boundary. Particularly, the uniform distribution of Cu element in both the Li-CuMH SSE and LFP laminate confirms the successful embedding of the Li-CuMH into the LFP cathode (Fig. 5c).

The rate performance of the LFP/Li-CuMH/Li cell was estimated by galvanostatic tests at room temperature. As manifested in Fig. 5d and S34, the LFP/Li-CuMH/Li cell can well operate in a potential range of 2.5 to 3.9 V at various rates up to 1 C (1 C = 170 mAh g⁻¹), concurrently achieving specific discharge capacities of 156.4 and 134.4 mAh g⁻¹ at 0.1 and 0.5 C, respectively. Moreover, when the specific current is

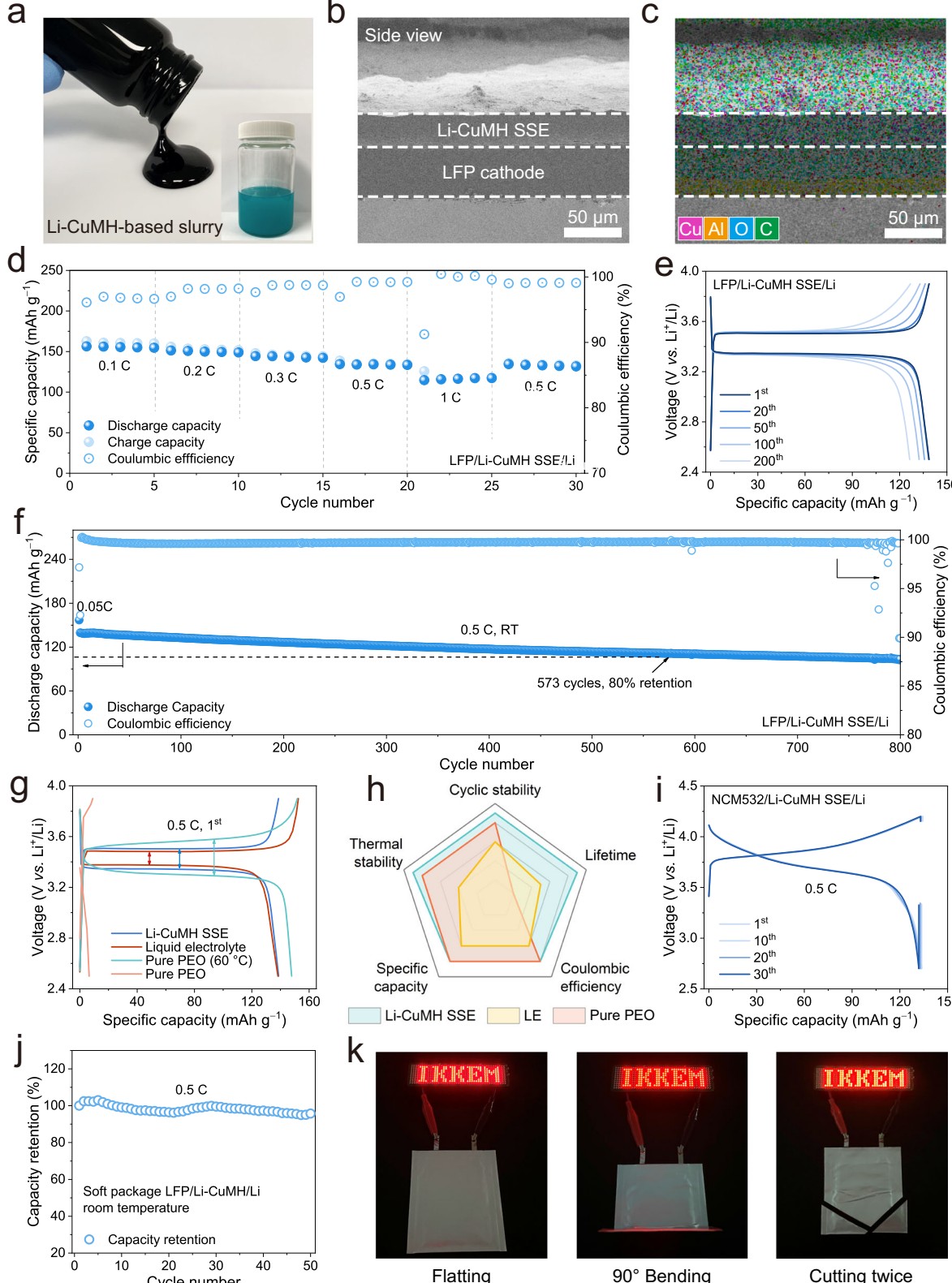

**Fig. 5 | Li-CuMH SSE-based solid-state full batteries. a** Digital photographs of the LiFePO₄ (LFP) cathode slurry containing the Li-CuMH emulsion (inset). **b** Side view SEM image and **c** the corresponding EDS mapping of the pressed Li-CuMH SSE onto LFP laminate. The colors of Cu, Al, O and C elements are pink, yellow, blue and green, respectively. **d** Rate performance of the LFP/Li-CuMH SSE/Li battery at various current rates at 25 °C. **e** Charge/discharge profiles of the LFP/Li-CuMH SSE/Li battery after various cycles at 0.5 C (1 C = 170 mAh g⁻¹) and room temperature. **f** Long-term cycling stability of the LFP/Li-CuMH/Li battery at 0.5 C at 25 °C. **g** First-cycle charge/discharge profiles of LFP/Li full batteries based on three types of electrolytes at 0.5 C. **h** Radar plot of comprehensive performance of LFP/Li batteries with different electrolytes. **i** Charge/discharge profiles of the NCM532/Li-CuMH SSE/Li full battery after various cycles. **j** Cycling stability of soft-package LFP/Li-CuMH SSE/Li cell at 25 °C. **k** Working states of the soft-package LFP/Li-CuMH SSE/Li batteries under various harsh conditions including flatting, bending and cutting.

switched back to 0.5 C, the reversible capacity basically remains unchanged, indicating the high tolerance of the LFP/Li-CuMH/Li cell under high power conditions. The long-term cycling stability of the LFP/Li-CuMH/Li cell at 0.5 C was further evaluated at room temperature (Figs. 5e and 5f). Encouragingly, the LFP/Li-CuMH/Li exhibits a high initial capacity of 139.5 mAh g$^{-1}$ and a capacity retention of 80% capacity is attainable after 573 galvanostatic charge/discharge cycles. This exceeds the requirement of maintaining 80% capacity after 500 cycles in practical application. The cycling stability can be indirectly confirmed by the ex situ EIS tests (Fig. S35). The Nyquist plots, especially $R_{SEI}$, show negligible changes after various cycles, revealing the high stability of the interface and the compatibility of Li-CuMH SSE with positive/negative electrodes. For comparison, full cells with liquid organic electrolyte and solid polymer PEO electrolyte were assembled. Figure 5g provides the first-cycle charge/discharge profiles of LFP/Li full cells based on three types of electrolytes at 0.5 C. Evidently, the LFP/Li full cell based on the PEO electrolyte cannot work at room temperature owing to its poor ionic conductivity. Notably, the cell can only exhibit good performance at 60 °C, but still suffers from large polarization and causes a short circuit after 50 cycles (Fig. S36). By contrast, the LFP/Li full cell with the liquid electrolyte exhibits the lowest polarization but the lowest initial Coulombic efficiency. However, a rapid capacity fading can be clearly observed in Fig. S37, in which the capacity retention decreases to 80% after only 200 cycles, demonstrating the incompatibility of the 1 M LiTFSI in DOL/DME electrolyte with electrodes. CV measurements were conducted to figure out the reaction reversibility and reaction activity of LFP/Li batteries based on the Li-CuMH SSE and the liquid electrolyte at room temperature (Fig. S38). A pair of redox peaks appear in both two cells during the first cycle, corresponding to the delithiation/lithiation process of LFP. During the following cycles, the peak current of the liquid electrolyte-based cell decreases with the cycle, while the peak current of LFP/Li-CuMH SSE/Li cell increases conversely, indicating the increasing reaction kinetics of the latter.

Figure 5h and Table S5 comprehensively summarize the performance properties of full cells with the Li-CuMH SSE, liquid electrolyte, and solid polymer PEO electrolyte, and compare them in the radar chart. Apparently, the Li-CuMH SSE-based full cell shows a full range of exceptional advantages including cycling stability, thermal stability, specific capacity and etc. The cycling performance of LFP/Li cells with various types of SSEs in the reported results is also compared in Table S6. LFP/Li-CuMH SSE/Li exhibited a superior cycling stability at high specific current and room temperature. Moreover, benefiting from the wide working potential window of 0-4.7 V, the Li-CuMH SSE is well compatible with high-voltage LiNi$_{0.5}$Co$_{0.2}$Mn$_{0.3}$O$_2$ (NCM532) cathode materials (Fig. 5i). The as-constructed NCM532/Li-CuMH SSE/Li full cell can operate in a potential range of 2.7–4.2 V at room temperature and delivers a reversible capacity of 132.8 mAh g$^{-1}$ at 0.5 C.

As an application demonstration, soft-package LFP/Li-CuMH SSE/Li batteries were assembled and estimated at room temperature. As given in Fig. 5j, the capacity of the soft-package cell remains basically stable with only small fluctuations up to 50 cycles, revealing the commercial application potential of the Li-CuMH SSE. To further evaluate the safety and reliability of the soft-package LFP/Li-CuMH SSE/Li batteries, harsh conditions such as bending and cutting were adopted subsequently. As shown in Fig. 5k, the soft-package batteries can light up an LED under normal conditions, and the light intensity can be perfectly maintained even after bending 90° and cutting twice.

## Li-CuMH SSE optimizes the interfacial SEI layer

The favorable compatibility of the Li-CuMH SSE with Li metal anodes has been fully demonstrated by the electrochemical measurements. To further clarify the influence of the Li-CuMH SSE on Li metal, the morphology of Li anodes disassembled after long-term cycling from Li/Li symmetric cells with liquid electrolyte and Li-CuMH SSE under the

same plating conditions was compared in detail. As displayed in Fig. S39, the Li metal surface presents a smooth morphology without obvious cracks and dendrites after cycling in the Li/Li-CuMH SSE/Li cell for 200 h. In stark contrast, severe dendrite growth can be observed on the Li metal surface after cycling in a liquid electrolyte environment. The stability of Li metal anode can be further clarified from the perspective of constructing a stable SEI film[63,64]. Atomic force microscopy (AFM) tests were conducted in an inert atmosphere to map the topography of Li metal surface and quantitatively measure the mechanical strength of the SEI layer (Fig. 6a and Fig. S40). As provided in Fig. 6a, the 3D AFM image of the SEI layer on the Li anode in the Li/Li-CuMH SSE/Li cell shows a continuous smooth surface without obvious fluctuations. The Young's elastic modulus derived from the corresponding force-separation curve of the SEI layer is 0.87 GPa (Fig. 6b). By contrast, the 3D AFM image of the SEI layer on the Li anode in the liquid electrolyte demonstrates a highly undulating surface morphology, concurrently exhibiting a low Young's elastic modulus of only 0.19 GPa (Fig. S41). Apparently, the Li-CuMH SSE plays an important role in the formation of a compact, continuous and high-strength SEI film owing to its mechanical rigidity.

In order to deeply understand the influence of Li-CuMH SSE on the composition of SEI film (i.e., interaction between the Li-CuMH and Li anode), the chemical composition of the SEI layer after 50 cycles was investigated by in-depth X-ray photoelectron spectroscopy (XPS). As shown in Fig. 6c, S42 and S43, the composition distribution at different depths in the SEI layer is shown on the F 1s, N 1s and C 1s spectra. Evidently, two peaks located at 688.8 and 684.6 eV in the F 1s spectra correspond to C-F and Li-F, respectively, which are attributed to the decomposition of LiTFSI salts. Notably, the peak intensity of Li-F is stronger than C-F in the outer layer of SEI on the Li metal surface taken from the Li/Li-CuMH SSE/Li cell. However, for the SEI layer on the Li metal surface after cycling in the liquid electrolyte (LE), the opposite is true. Fig. S44 further quantitatively provides the proportion of different components in the surface SEI film. The N 1s validated that the strength of Li$_3$N in Li-CuMH SSE was higher than LE not only in the outer layer but also in the inner layer of SEI (close to the lithium metal). Li-F and Li$_3$N are known to be fine ionic conductors and electron insulators, which can rapidly transport Li$^+$ ions to the Li metal surface and inhibit electrons from crossing the interface[62,65,66]. Thus, the rich composition of Li-F and Li$_3$N helps to build a more uniform, compact and thin SEI layer, which ultimately contributes to the uniform Li deposition (Fig. 6f). On the contrary, the electron-rich C-F bond is strongly polarized and is usually regarded as the diffusion barrier of Li$^+$ ions, which can easily cause uneven Li deposition and dendrite growth (Fig. 6g)[61].

Moreover, as shown in C 1s spectra, four distinct peaks located at 284.8, 286.5, 289.7 and 292.9 eV are assigned to C-C, C-O, CO$_3^{2-}$ and C-F groups, respectively, indicating the coexistence of organic and inorganic products in the SEI layer. Notably, the signal of CO$_3^{2-}$ represents the inorganic species of Li$_2$CO$_3$, which is not conducive to the formation of a dense and stable SEI layer due to its low Li$^+$-ion conductivity ($\sim$10$^{-8}$ S cm$^{-1}$)[67]. Apparently, the intensity of CO$_3^{2-}$ of the SEI layer on the Li metal surface taken from the Li/Li-CuMH SSE/Li cell is significantly lower than that of Li metal after cycling in the organic electrolyte at each depth (Fig. 6c and S42). Moreover, a new peak of C-SO$_X$ appears in the inner layer of SEI layer after cycling in the liquid electrolyte, suggesting a more intense decomposition of TFSI$^-$ on the Li metal surface. This may result in a non-uniform SEI film with a fragile multilayer structure (Fig. 6g)[68]. To further elucidate the structure and composition of SEI layers, time-of-flight secondary ion mass spectroscopy (TOF-SIMS) analysis was performed (Fig. S45). Figures 6d, 6e and S46 depict the 3D rendering profiles of the SEI on the Li metals disassembled from cells with the Li-CuMH SSE and liquid electrolyte after long-term cycling, in which the relative content and spatial distribution of the main decomposed ion fragments such as C$_2$H$_2$O$^-$, LiF$_2^-$ and LiF$^-$ are provided. The high intensity of LiF$_2^-$ and LiF$^-$ fragments again

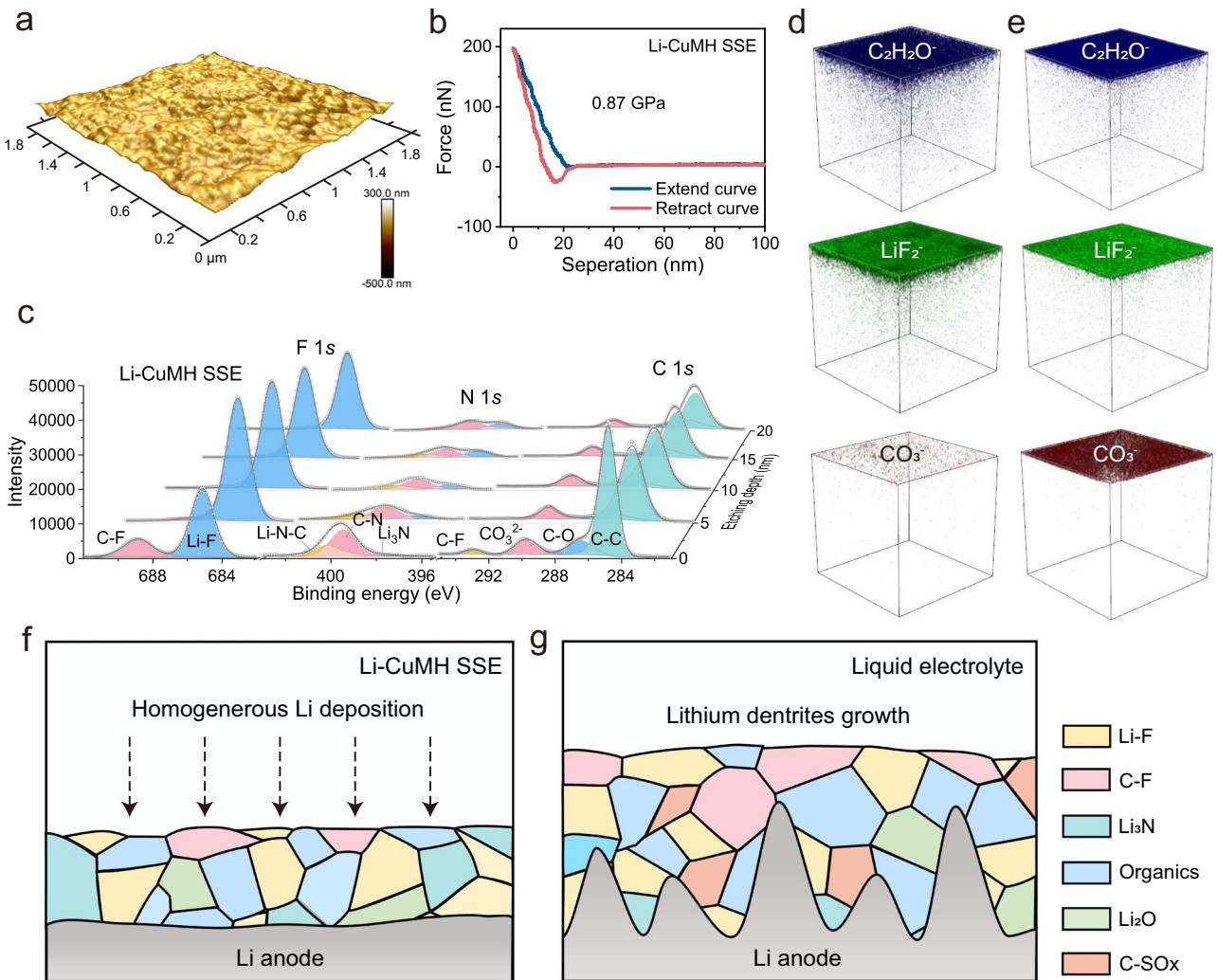

**Fig. 6 | Optimized SEI layer on the Li metal surface. a** 3D AFM image and **b** the corresponding force-separation curve of Li anode disassembled from Li/Li symmetric cell with Li-CuMH SSE. **c** F 1$s$, N 1$s$ and C 1$s$ in-depth XPS spectra of SEI layers on Li anodes disassembled from Li/Li symmetric cells with Li-CuMH SSE. The raw data and fitted data plots are shown as red dotted lines and grey solid lines, respectively. **d, e** TOF-SIMS 3D reconstruction of $LiF_2^-$, $C_2H_2O^-$ and $CO_3^-$ species of SEI layers on Li anodes disassembled from Li/Li symmetric cells with Li-CuMH SSE and liquid electrolyte. **f, g** Schematics of SEI composition affected by Li-CuMH SSE and liquid electrolyte.

confirms the LiF-rich SEI layer on the Li metal surface in the Li-CuMH SSE-based cell. Moreover, all SEI films exhibit a gradient hierarchy of decreasing $C_2H_2O^-$ content from top to bottom, which is in good accordance with the distribution of C-C signal in the XPS results. Note that the SEI layer on the Li metal after cycling in the liquid electrolyte shows stronger $SO_2^-$ and $CO_3^-$ intensity, in line well with the XPS results (Fig. S43). The in-depth XPS and TOF-SIMS measurements have verified that the Li-CuMH SSE can suppress the electrolyte decomposition and promote the generation of $Li^+$ conductors, *e.g.*, LiF, $Li_3N$ (Fig. 6f). The obtained SEI layer with low $SO_2^-$ and $CO_3^-$ content is conducive to reduce the resistance of $Li^+$ ions transport at the interface and eventually contributes to the highly reversible Li striping/plating process[1,69].

In summary, we have presented our findings on the rational design, fabrication, and in-depth understanding of a previously unexplored 2D lamellar SSE of Li-CuMH. This SSE exhibits promising ionic conductivity ($1.17 \times 10^{-4}$ S $cm^{-1}$), low activation energy (0.123 eV), and a high $Li^+$ transference number (0.77) at room temperature, owing to its highly crystalline structure with well-defined, angstrom-scale ion channels. The rapid ion-hopping mechanism of $Li^+$ ions is facilitated by the abundance of electronegative oxygen-containing functional groups and structural water, as revealed by experimental

investigations and theoretical calculations. Furthermore, due to its favorable attributes of electrochemical stability, flexibility, and flame retardance, the assembled solid-state LFP/Li-CuMH SSE/Li batteries achieve superior specific capacity and prominent long-term durability of over 800 cycles compared to solid PEO electrolyte and organic liquid electrolyte systems. This work paves the way for exploring more self-assembled solid-state electrolytes and their application in high-efficiency, stable, and long-cycling solid-state lithium metal batteries.

## Methods
### Synthesis of copper maleate monohydrate (CuMH) nanoflakes
Copper maleate monohydrate (CuMH) nanoflakes with controllable 2D morphology were prepared by the following procedure. Briefly, a fresh Cu foil (99.999% purity) was directly placed in an acetonitrile (AN, Guoyao, GR) solution dissolved in 6 mM maleic acid (MA, TCI, 99.0% purity) and 0.36 mM lithium nitrate ($LiNO_3$) under atmospheric environment. During standing for 24 hours, the blue precipitate was continuously deposited in the reaction container. After that, the precipitates were collected by centrifugation, washed with AN solvent to remove residual MA and $LiNO_3$ for three times and vacuum dried at 60 °C for 24 hours prior to use.

## Fabrication of flexible CuMH films

The flexible and compact CuMH film were fabricated by rolling a viscous paste comprising CuMH powders and polytetrafluoroethylene (PTFE, 60 wt.% in $H_2O$, Macklin) emulsion in a solid ratio of 9:1. The viscous paste was thoroughly ground until CuMH powders and PTFE binder were completely and evenly mixed, and a small amount of ethanol was supplemented to ensure the rheology of the paste. Subsequently, the CuMH-PTFE paste was fed into a roller press machine (MTI Technology Co., Ltd.) and repeatedly rolled. The obtained CuMH films were firstly dried at 60 °C for 10 min to evaporate the residual ethanol and then punched into discs (19 mm in diameter). The CuMH discs were then vacuum dried at 100 °C for 48 h and transferred into a glovebox prior to use.

## Preparation of the Li-CuMH SSE

The as-prepared CuMH films were soaked into the 1 M LiTFSI −1, 3-dioxolane /1, 2-dimethoxyethane (DOL/DME, 1:1, v/v) electrolyte at 60 °C for at least 24 h to ensure that Li+ ions fully penetrate into the regular channels of CuMH. Then, the soaked membranes were removed from the liquid electrolyte, cooled to room temperature, and washed with the DME solvent to remove the adsorbed LiTFSI salts. These procedures were repeated at least three times to achieve full implantation of Li+ ions. After sufficient impregnation, the residual solvent on the CuMH film was removed by pressing and wiping with the dust-free cloth. The obtained Li-CuMH SSE films were then dried at 60 °C for 2 h and then vacuum dried for 2 h at the same temperature. The above two steps were repeated until all solvents and adsorbed water were thoroughly removed from the Li-CuMH SSE film (except for structural water). Note that all obtained Li-CuMH SSEs were aged for at least 3 h prior to battery assembly.

## Preparation of the metal electrode

The Li metal foils were purchased from China Energy Lithium Co., Ltd. (99.9% purity) and stored in the glovebox. The thickness and diameter of Li foils for CR2032-type batteries are 1 mm and 16 mm, respectively. The area and thickness of Li foils for soft-package batteries are about 26.1 cm² (45×58 mm) and 100 μm, respectively.

## Cathode electrode preparation

LiFePO$_4$ (LFP) and LiNi$_{0.5}$Co$_{0.3}$Mn$_{0.2}$O$_2$ (NCM532) materials, Super P carbon blacks (SP) and polyvinylidene (PVDF) binder were purchased from MTI Technology Co., Ltd. The LFP and NCM532 cathode electrodes were prepared with the participation of Li-CuMH powers to guarantee an efficient ion-conducting network. Briefly, Li-CuMH powers were first dispersed into the N-Methylpyrrolidone (NMP) solution to obtain a homogeneous 5 wt.% emulsion. The LFP or NCM cathode laminates were fabricated by casting the NMP-based slurry comprising active materials, SP, PVDF and Li-CuMH onto the Al foil in a mass ratio of LFP/NCM: SP: PVDF: Li-CuMH= 7:1:1:1. The slurry was thoroughly mixed using a twin-shaft ball mixer (Thinky AR 100) at 2000 rpm min⁻¹ for 30 min. The obtained LFP cathode laminates were pounced into discs (11 mm in diameter, 44 μm in thickness) and vacuum dried at 80 °C for 12 h, and the active materials mass loading was around 4 mg cm⁻². The obtained NCM cathode laminates were pounced into discs (11 mm in diameter, 28 μm in thickness) and vacuum dried at 80 °C for 12 h, and the active materials mass loading was around 3 mg cm⁻².

## Battery assembly and electrochemical measurements

All electrochemical measurements were performed using CR2032 batteries assembled in an argon-filled glove box, with $H_2O$ and $O_2$ content less than 0.01 ppm. And all electrochemical energy storage tests are carried out in the incubator (SPX-150BIII, TAISITE INSTRUMENT). ionic conductivity of the Li-CuMH SSE was measured through electrochemical impedance spectroscopy (EIS) tests by constructing an SS/Li-CuMH SSE/SS (SS = stainless steel) battery. The temperature and frequency were varied in the range of 25 to 80 °C and $10^{-2}$ to $10^6$ Hz, respectively. The ionic conductivity, σ, can be evaluated using the equation below:

$$\sigma = \frac{L}{R_b S} \tag{1}$$

where $R_b$, $L$ and $S$ represent the bulk resistance, the thickness of SSE and the area of SS electrode, respectively. The Li+ transference number ($t_{Li^+}$) was measured on Li/Li-CuMH SSE/Li symmetric cells by the chronoamperometry test. $t_{Li^+}$ can be evaluated using the following equation:

$$t_{Li^+} = \frac{I_{SS}}{I_O} \times \frac{V - I_O R_O}{V - I_{SS} R_{SS}} \tag{2}$$

where $V$, $I_O$, $I_{SS}$, $R_O$, $R_{SS}$ represent the applied voltage (10 mV), the initial and steady-state currents and the impedance before and after polarization, respectively. The electrochemical stability window (ESW) of the Li-CuMH SSE was determined by a Li/Li-CuMH SSE/SS battery. The linear sweep voltammetry (LSV) test was conducted to evaluate the high-voltage resistance of the Li-CuMH SSE in a potential range of 3 to 6 V, and cyclic voltammetry (CV) profiles were recorded to estimate the lower limit of the operating voltage between −0.5 to 4 V at a scan rate of 0.1 mV s⁻¹. The stability of the Li-CuMH SSE against Li metal electrode was estimated on symmetric Li/Li-CuMH SSE/Li cells at various current densities and areal capacities at room temperature. The galvanostatic electrochemical measurements of solid-state batteries were conducted on LANHE CT2001A battery testing system (LAND Electronics Co., China). The LFP/Li-CuMH SSE/Li coin cells and pouch cells were operated between 2.5 and 3.9 V, and the NCM/Li-CuMH SSE/Li coin cells were cycled between 2.7 and 4.2 V at room temperature, respectively. Note that the full batteries were activated at 0.05 C for several cycles. The specific power values for all batteries were calculated by the following equation.

$$P = \frac{U \times I}{M_a} \tag{3}$$

where $U$, $I$ and $M_a$ represent the output voltage and output current of batteries, and the mass loading of active materials. For comparison, LFP/Li full cells based on the liquid electrolyte (1 M LiTFSI in DOL/DME with 2 wt% LiNO$_3$) and all-solid-state PEO electrolyte (EO:Li = 10:1) were assembled and measured. Note that the liquid electrolyte-based and PEO-based full cells were operated at room temperature and 60 °C, respectively. All batteries were left standing for 6 h before various electrochemical measurements.

## Materials characterization

The morphology and element distribution were characterized with a Field emission scanning electron microscopy (FESEM, Zeiss), equipped with an EDS probe. The microstructure diffraction pattern was collected by X-ray diffraction (XRD, Ultima-IV) at a scan rate of 10° s⁻¹, and nanostructure scattering pattern was recorded by small angle X-ray scattering (Nanostar U SAXS, Bruker) from 3 nm⁻¹ to 35 nm⁻¹. Single-crystal X-ray diffraction pattern was collected on XtaLAB Synergy (Dualflex; HyPix) and XtaLAB Synergy R (DW system; HyPix) diffractometers using Cu Kα (λ = 1.54184 Å) micro-focus X-ray sources (PhotonJet (Cu) X-ray source). The CrysAlisPro software was used to collect and reduce the raw data. The structures were solved by ShelXT with intrinsic phasing and refined on $F^2$ using full-matrix least-squares methods, with ShelXL and Olex2 used as graphical user interfaces. The depth-resolution etching X-ray photoelectron spectroscopy (XPS) spectra were performed by XPS (Thermo Fisher Scientific) with an etching depth of 5 nm per layer, applying an Al Kα radiation source

(1486.6 eV) and under ultra-high vacuum. The chemical bonding information was analyzed by Fourier transform infrared spectroscopy (FT-IR, Bruker Optics, ATR mode) and laser confocal Raman microscope (Nanophoton, Vis-NIR-XU) from 500 to 4000 cm$^{-1}$. The $^7$Li environment of solid samples were collected on a solid-state nuclear magnetic resonance (ss-NMR, 400 MHz Bruker Avance). Thermogravimetric analysis (TGA) was performed on a SDTQ600 analyzer ranging from 30 to 800 °C at a heating rate of 10 °C min$^{-1}$. The liquid $^1$H NMR measurements were conducted by AVANCE NEO 500 MHz Digital FT-NMR Spectrometer (NMR, AVANCE NEO 500 M Hz, BRUKER). Specifically, the CuMH-based samples were prepared by dissolving them in the deuterated dimethyl sulfoxide (DMSO-d6, 99.9%, Energy Chemical) under controlled heating and stirring conditions. The depth composition distribution was measured by time-of-flight secondary ion mass spectrometry (TOF-SIMS, IONTOF M6). Atomic force microscopy (AFM) tests were performed on a Model 5500 scanning probe microscope in an argon-filled glove box. AFM imaging were recorded through contact mode, and the force constant measured by force curve was 10 N m$^{-1}$.

## Computational methods

Spin-polarized Density Functional Theory (DFT) calculations were performed on the Vienna Ab initio Simulation Package (VASP.6.1)[70,71], using the generalized gradient approximation of exchange (GGA) proposed by Perdew, Burke, and Ernzerhof[72]. The projector-enhanced wave (PAW)[73] method was used to handle interaction between valence electrons and ions. For the optimization of Cu$^{I/II}$MH and Li-Cu$^{I/II}$MH crystal structures, the kinetic energy cutoff of plane-wave expansion was set to 520 eV and the $2 \times 2 \times 4$ k-point sampling[74] was employed. The convergence criteria for the total energy and force were less than $1.0 \times 10^{-5}$ eV and 0.01 eV Å$^{-1}$, respectively. The MD simulations were performed on a $2 \times 3 \times 2$ supercell at temperatures of 350, 450, 550, 650 and 750 K using a NVT ensemble[75]. The MD simulation was run for 8 ps with time step of 1 fs. A G-centered k-point mesh of $1 \times 1 \times 1$ was employed for MD calculations, ensuring a reasonable computational cost. The bond valence site energy (BVSE) calculations based on SoftBV software tool[76] were performed to investigate the diffusion pathways and the migration energy barriers of Li-ion in the Li-CuMH. The diffusion pathways of Li-ions were defined by the connected regions of low bond valence site energy. The global minimum value of $E_{BVSE(Li)}$ required to form a migration path was determined as the Li-ion migration energy barrier. Visualization of crystal structures and ion migration paths can be achieved by the VESTA program[77]. The electrostatic potential diagram is drawn using Multiwfn[78] in combination with VMD[79]. The electrostatic potential and orbital energies of CuMH and Li-CuMH molecular were calculated on Gaussian program, using the hybrid B3LYP functional and the basis set 6-311+G*. In order to correct for weak interactions, the D3 version of Grimme's dispersion with Becke–Johnson damping (GD3BJ) was adopted.

## Reporting summary

Further information on research design is available in the Nature Portfolio Reporting Summary linked to this article.

## Data availability

The data supporting this study have been included in the main text and Supplementary Information. Source data are provided with this paper. The other relevant data supporting this work are available from the corresponding authors on reasonable request. Source data are provided with this paper.

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

## Acknowledgements

This work was supported by the National Natural Science Foundation of China (Grant Nos. 52122211-Q.Z., 92372101-L.Z., 52072323-Q.Z. and 21875155-L.Z.), the support of National Key Research and Development Program of China (2021YFA1201502-L.Z.), the Frontier Exploration Projects of Longmen Laboratory (Grant No. LMQYTSKT008-Q.Z.), the Shenzhen Technical Plan Project (No. JCYJ20220818101003008-Q.Z.) and the "Double-First Class" Foundation of Materials and Intelligent Manufacturing Discipline of Xiamen University. L. Zhang and Q.B. Zhang acknowledge the support of Nanqiang Young Top-notch Talent Fellowship in Xiamen University. X. Zhan, M. Li, X.L. Zhao and Y.N. Wang contributed equally to this work.

## Author contributions

L. Zhang and Q. B. Zhang designed and supervised this work. X. Zhan and M. Li carried out the synthesis and the electrochemical experiments. M. Li and X. Zhan performed PXRD, FT-IR, Raman, SEM, TG measurements, flammability tests and result analysis. Y.N. Wang, X.L. Zhao and J.D. Lin carried out DFT, BVSE and AIMD calculations. W.W. Wang and J.W. Yan carried out the AFM measurements. Z.A. Nan performed the SXRD experiment. M. Li, X. Zhan and X.L. Zhao wrote this paper. L. Zhang, Q.B. Zhang and J. J. Liu appraised and revised the paper. M. Li, X. Zhan, X.L. Zhao, Y.N. Wang, S. Li, W.W. Wang, J.D. Lin, Z.A. Nan, J.W. Yan, Z.F. Sun, H.D. Liu, F. Wang, J.Y. Wan, J.J. Liu, Q.B. Zhang, and L. Zhang participated in the analysis of experimental data and the discussion of results, as well as the preparation of manuscript. We thank Prof. Y.G. Xiao from Peking University, Shenzhen Graduate School, for his valuable discussion.

## Competing interests

The authors declare no competing interests.

## Additional information

[1]State Key Laboratory of Physical Chemistry of Solid Surfaces, College of Chemistry and Chemical Engineering, College of Materials, Tan Kah Kee Innovation Laboratory, Collaborative Innovation Center of Chemistry for Energy Materials, Xiamen University, Xiamen 361005, Fujian, China. [2]State Key Laboratory of High Performance Ceramics and Superfine Microstructures, Shanghai Institute of Ceramics, Chinese Academy of Sciences Shanghai 200050, China. [3]Chemical Engineering, UC San Diego La Jolla, CA 92093, USA. [4]Department of Materials Science, Fudan University Shanghai 200433, China. [5]Future Battery Research Center, Global Institute of Future Technology, Shanghai Jiaotong University Shanghai 200240, China. [6]Shenzhen Research Institute of Xiamen University Shenzhen 518000, China. ✉e-mail: jliu@mail.sic.ac.cn; zhangqiaobao@xmu.edu.cn; zhangli81@xmu.edu.cn

