## [Peer Review File · Nature Communications]

REVIEWER COMMENTS

Reviewer #1 (Remarks to the Author):

The report from Li and coworkers presents a hybrid liquid/organic-inorganic-crystalline electrolyte that shows good cycling properties and a high voltage stability window. The activation barriers obtained from Arrhenius analysis suggest low barriers to ion migration, though the actual conductivity is middling (10^{-4} S/cm). The lithium ion transference number is excellent at 0.77.

Despite the promising electrochemical characterization of this electrolyte, the authors do not offer convincing evidence that lithium is intercalated into the crystals of the bulk, nor the through-channel conductivity that the authors simulate and claim to be the mechanism. In fact, some of the characterization evidence suggests the opposite: that lithium ions do not fill the interlayers. While the mechanistic investigation through computation is compelling, and the electrochemical data is impressive, a big missing piece is evidence that lithium intercalation is relevant, and this is not lithium percolation through a percolating network.

A big problem with the proposed mode of conductivity is that the assumption is that LiTFSI dissociates and the lithiums move en masse into the crystal, and leave the TFSI⁻ anions behind. This is highly improbable. It would lead to an immense positive charge buildup in the interior of the crystal, and an immense negative charge buildup outside the crystal. It is difficult to separate even one ion pair. The notion that millimoles of ions would spontaneously separate just by "soaking" in a liquid electrolyte seems very unlikely without an immense driving force. There is no clear driving force for such a massive charge separation on bulk scale.

The SAXS pattern shows no change after Li loading. This strongly suggests there is no Li⁺ intercalation. The inclusion of Li into the lattice should swell the crystal (we all know that cathodes and anodes swell when they are loaded with Li⁺. Why wouldn't this material?), and alter the XRD pattern, and diffraction angle of the reflection peaks. The SAXS peaks do not change upon lithiation, suggesting there is no change in the unit cell, and no lithium uptake.

The authors note a difference in the SAXS peaks, but their observation is problematic:

"Note that the weakening of the intensity of all peaks represents a decrease in the electron density of Li-CuMH units, which could likely be attributed to the possible interaction between Cu (II) and TFSI⁻ anions[30]."

This is not really how one would expect an XRD to change in response to an alteration of composition. Changes in atomic structure should change the peak positions of the XRD pattern. Even in the unlikely event the unit cell is not perturbed, but new atoms are inserted, a change in atomic composition will increase the intensity of some peaks (that have stronger or more scatterers on those hkl planes) and decrease the intensity of other peaks. The authors argue that a uniform decrease in all peak intensities could be meaningful, when the only reasonable interpretation for an identical but weaker pattern, is that less sample was used.

The authors base much of the arguments and characterization on a single crystal structure, but this data is not provided. The authors must include a CIF so that reviewers can evaluate the single-crystal structure.

In Figure S8, the authors argue that the appearance of a few new peaks suggests Li intercalation, but this is also unsubstantiated. If a crystal structure changes, one does not expect to get new XRD peaks in addition to the old ones. The whole pattern must change. A PXRD is not like an NMR where you get new signals for new atoms. It is a Fourier transform of the electron density map. Everything in the XRD pattern changes drastically if you perturb the periodic lattice.

The authors also use NMR broadening arguments to support the intercalation of Li but these are not convincing either. These results would be expected whether the Li was in the solid grains, or in the boundaries between.

The authors offer ESP mapping as a surrogate evidence for lithium intercalation, but this is not useful. Of course the oxygenic ligand sites are more negative and *would* have a higher affinity for Li⁺ than the uncharged atoms, but this is not evidence of LiTFSI mass dissociation and intercalation into a crystal without lattice disruption.

The migration paths shown in Figure 2g offer a testable hypothesis for the expected XRD changes in response to Li intercalation. If lithiums do indeed slip into this region without expanding the crystal, you are inserting ions (and hence electron density) into the 200 plane. If this hypothesis is correct, this increase in electron density should result in an increase in the intensity of the 200 reflection due to the additional scattering from the new Li ions, and a decrease in the intensity of 100 because the new Li ions are out of phase with the Cu atoms on the 100 plane, and give some destructive interference. But the SAXS seems to suggest there is no change in the intensities (or positions) of the XRD peaks at all.

The simulated models of ion diffusion represent an opportunity to test the assumption that the crystal lattice does not swell upon lithium intercalation. In the models of lithium ion channels, is the simulated model allowed to swell to accommodate the Li ions? Does the lithium actually fit here without expanding the lattice? What are the O---Li distances? Are they realistic? Does this AIMD simulation predict a swelling of the crystal lattice? If so, the authors should be able to approximate the new cell parameters that would be expected for Li intercalation from their simulation.

Minor problems/suggestions

The authors refer to the local electronegativity of COO⁻ and H₂O, which is improper. Groups/molecules don't have electronegativities; atoms do. The authors mean to say that the ligands are negatively charged as expected due to the electronegativity of the oxygen atoms.

Li/Li symmetric cell is not a "battery" strictly speaking, because it cannot spontaneously charge and discharge. Please refer to the Li|SSE|Li system as a "symmetric cell" instead of a battery

Cycling with increasing voltage until failure (Figure 4a) strikes me as a strange approach to test the cycling limits of the electrolyte. Dendritic failure comes from either high current or high cycle number, and this experiment is changing them both at once, and then attributing the failure to the voltage only, and not the cycle number. Is this a common practice? I'm unfamiliar with it. If one cycles the symmetric cell at 0.8 V does it really hold up well?

The authors conclude that the electrolyte has good interfacial properties with the electrodes, but I am concerned it's actually a slurry with residual liquid electrolyte. The authors prepared the electrolytes by soaking in liquid electrolyte, rinsing with DMF, and then padding them dry, and evacuating. But these solvents are very low-vapor pressure. I would suspect this is a liquid electrolyte percolating between the particles. The particles are probably not directly contacting the electrode, which explains why the SEI analysis turns up only components of the anion (F, S, C, O), and not the discriminating component of the electrolyte: copper. An SEI formed from the solid electrolyte should contain copper atoms.

In summary, the authors do appear to have discovered something interesting. The electrochemical data shows that this electrolyte works well and is very promising. However, for *Nature Commun.*, a much more well-developed characterization of the active material, and better supported claims are needed. If the authors can demonstrate convincingly with data (not just computation) that there is lithium in the crystal lattice, this paper could be reconsidered. However, if this effect is a lithium intercalation along a percolating network (e.g., along the surface of the crystals) as I suspect, this could still be a useful technology, but is less novel in that it is similar to the addition of nanophases to polymer electrolytes, which generates percolating surface networks of mobile ions, and in this case, should be submitted elsewhere once the mechanism and mode of ion transport is better supported.

Reviewer #2 (Remarks to the Author):

In the manuscript titled "Self-assembled hydrated copper coordination 1 compounds as ionic-conductors for room temperature solid-state batteries"; Zhang and his colleagues have successfully integrated the characteristics of inorganic solid-state electrolytes, specifically the Li diffusion channel, with the coupling dissociation phenomenon of polar organic solid-state polymer electrolytes. This integration has led to enhanced Li transportation in a 2D lithiated Cu-maleate coordination compound, which serves as an organic-ionorganic hybrid. The compound exhibits a remarkable breakthrough in Li conductivity, as demonstrated by its exceptional performance in a Li/Li-CuMH SSE/Li symmetric cell and in an LFP/Li-CuMH/Li full cell.

The manuscript presents a meticulous investigation of the plausible mechanism, supported by both computational modeling and experimental evidence. The methodical use of graphical and pictorial representations effectively conveys the findings. The manuscript is well-organized, with a wealth of data and a profound understanding of the subject matter, making it highly suitable for acceptance in *Nature Communications* after revision.

Furthermore, the recommended suggestions to refine and enhance the transparency of the manuscript will likely improve its suitability for publication.

Some scientific suggestions:

1. In the manuscript, the authors initially mention that CuMH exhibits a 2D nanoflake structure, as confirmed by SEM. Considering the lithiation process involved in the formation of Li-CuMH SSE, it is essential to determine whether the nanoflake morphology is retained. If so, it is crucial to clearly explain the advantages of such a morphology in facilitating the conduction of Li-ions in bulk Li-CuMH SSE.

2. In order to determine the state of the crystallized water between the layers of Cu-MH during the lithiation process for Li-CuMH SSE production, it is recommended to perform TGA-DTA analysis and ¹H-Proton NMR verification.

Concern related to the contribution from proton conductivity from the structural water

(i) When considering the presence of water in the Li-CuMH SSE, as indicated by the single crystal structure of CuMH, it is important for the authors to address how they confirm that H₂O does not play a major role in proton conduction. There is a possibility of proton conduction being predominant, as protons are more mobile than Li ions. Alternatively, a dual effect of proton conduction and Li conduction could occur. This greatly depends on the Li/H₂O ratio within the overall solid electrolyte. To provide clarification, it is essential for the authors to provide the specific Li/H₂O ratio in the Li-CuMH SSE during the study.

(ii) The use of MSD (Mean Square Displacement) is an excellent technique to assess the migration rates of different constituents within a structure. While the authors calculated the MSD for COO⁻, HC=CH, and Cu(II) in Li-CuMH, it is important for them to also include the calculation for structural H₂O. This analysis would provide further insight into whether the observed partial contribution is attributed to proton conductivity or other factors."

(iii) (a) It would be beneficial for the authors to calculate the same ESP (Electrostatic Potential) mapping on the exact building unit of CuMH. This approach would enhance the prominence of the analysis and provide a better understanding of the basic character of the maleate unit and the structural H₂O. (b) In addition to the computational investigation, it is recommended that the authors complement their research with proton NMR investigation to assess the state of the structural H₂O during the Li transportation process. Performing the NMR analysis either ex situ or in situ would provide valuable insights. (c) To confirm the absence of a significant role of proton conductivity, it is crucial for the authors to conduct a control experiment using Li-CuMH and Cu-MH separately. The CuMH sample should not exhibit considerable ionic conductivity in the absence of Li.

3. The correlation between high voltage resistance and the low HOMO energy level of Li-CuMH should be compared with the theoretical oxidation potential (vs. Li/Li⁺) value derived from the HOMO energy level. Without a specific theoretically calculated oxidation value, it remains a gross approximation.

4. It would be helpful if the authors could explain how they calculated the transport number of lithium by combining the chronoamperometry and EIS techniques.

5. It is important to investigate the cause of sudden drops at 1.0 mA cm⁻² during galvanostatic cycling with step increased current densities to evaluate the durability of Li-CuMH SSE. Could it be attributed to complete destruction of the Li-CuMH SSE? Performing post-cycling experiments after applying the 1.0 mA cm⁻² current will provide valuable insights.

6. The gradual degradation mechanism of the synthesized solid-state electrolyte in the full cell or in the i/Li-CuMH SSE/Li configuration should be explained. The authors can investigate the gradual polarization of the Li-CuMH SSE or its swelling by performing ex situ EIS measurements after 500 cycles.

Some technical suggestions:

1. In some instances, the water molecules in hydrated Cu-MH are referred to as crystallized water, while in other places, they are called structural water. The author needs to use consistent terminology throughout the manuscript to avoid confusion. It is recommended to clarify and use a single term that accurately reflects the water molecules' coordination to the Cu centers as indicated by the crystallographic evidence.
2. To enhance technical clarity, it would be beneficial to include bond distances between the oxygen atoms from water-Cu and maleate-Cu centers. Additionally, a clear crystallographic figure illustrating the structural building unit of CuMH with all the coordination sites around the Cu center should be provided.
3. In the Results and Discussion section, it is advisable to create subheadings such as "CuMH synthesis" and "Li-CuMH SSE film preparation" to better represent the two steps involved in Li-CuMH SSE synthesis.
4. The solvent used for the emulsion to prepare the Li-CuMH film with PTFE should be mentioned.
5. The author should provide the CCDC (Cambridge Crystallographic Data Centre) number for the single crystal structure of CuMH.
6. Figure 1a does not clearly explain the synthetic procedure. It would be helpful to include additional information from the text in the figure. The incorporation of LiNO₃ should be explicitly mentioned. The presence of Li should also be identified in this figure.
7. It is recommended to replace Figure 1d with a new photograph that clearly shows both the unreacted Co-foil and the Li-CuMH coated portion in a single picture to provide a more realistic representation.
8. The claim of increased dissociation content and dissociation effect of LiTFSI in Li-CuMH based on the narrow Raman shift window by deconvolution is an overestimation. The author should avoid making such exaggerated claims.
9. In line 411, it is most likely referring to powder not power.
10. In line 419, it should be either "success of embedding" or "successful embedding."
11. The statement "Li-CuMH SSE optimizes the interfacial SEI layer" is a valuable inclusion. However, to maintain the readers' focus on the main topic and to prevent unnecessary elongation of the manuscript, it is suggested to move this portion to the supporting information.

Reviewer #3 (Remarks to the Author):

This article reports an interesting work on the development of SPE for solid state batteries. Before accepting, authors should revise the introduction section to include recent works, include more details on sample preparation, compare battery performance with the literature.

Point-by-point Response to Review Comments

We express our sincere gratitude to the reviewers for their insightful comments and valuable suggestions, which have greatly contributed to the improvement of this paper. We have thoroughly revised the manuscript based on these comments, making it more comprehensive, concise, and coherent. The reviewers' suggestions have also served as a valuable reminder for us to be meticulous in our scientific research. All the revisions have been incorporated into the revised manuscript. In order to address the reviewers' comments more clearly, we have categorized them into specific question areas and provided detailed responses as follows.

Reviewers' comments and our Response:

Reviewer #1 (Remarks to the Author): =====

The report from Li and coworkers presents a hybrid liquid/organic-inorganic-crystalline electrolyte that shows good cycling properties and a high voltage stability window. The activation barriers obtained from Arrhenius analysis suggest low barriers to ion migration, though the actual conductivity is middling (10^{-4} S/cm). The lithium ion transference number is excellent at 0.77.

Our response: We appreciate the reviewer's positive comments. The referee's encouraging comments have been carefully considered and thoroughly addressed, as shown below.

Q1: *Despite the promising electrochemical characterization of this electrolyte, the authors do not offer convincing evidence that lithium is intercalated into the crystals of the bulk, nor the through-channel conductivity that the authors simulate and claim to be the mechanism. In fact, some of the characterization evidence suggests the opposite: that lithium ions do not fill the interlayers. While the mechanistic investigation through computation is compelling, and the electrochemical data is impressive, a big missing piece is evidence that lithium intercalation is relevant, and this is not lithium percolation through a percolating network.*

Our response: We appreciate the reviewer for the very professional comment. This very wise question prompted us to rethink the Li-CuMH SSE from a microscopic point of view. We accordingly supplemented the related experiments to prove that Li^+ ions can indeed intercalate and fill the interlayers. Inspired by the reviewer's question, we first restudied the crystal

structures of as-prepared CuMH and Li-CuMH films. We found that in the original **Figure S8** of the supplementary information, the characteristic peak corresponding to the (001) crystal plane did change significantly after the sufficient Li⁺ ion implantation, mainly manifested as the peak position shifting negatively from 12.27° to 12.21°, accompanied by a decrease in peak intensity (**new Figure S13**). This clearly indicates that the crystal spacing indeed expands with the implantation of Li⁺ ions, and the diffraction intensity decreases accordingly.

New Figure S13

Figure S13. a) XRD spectra of CuMH film and Li-CuMH SSE film and b) the corresponding enlarged spectra in the 2-theta range of 11.5° and 13°.

On this basis, the crystal structures of Li-CuMH SSE films after soaking in the non-aqueous electrolyte containing LiTFSI salts for different times were further investigated. As shown in the **new Figure S14**, as the soaking time increases, the characteristic peaks located at around 12.3° continue to shift negatively, accompanied by a continuous decrease in peak intensity, indicating that Li⁺ ions are indeed constantly intercalated into crystal interlayers. This phenomenon is highly consistent with what we found in **Figure S13**. Electrochemical impedance spectroscopy (EIS) measurements were further carried out to estimate the change in ionic conductivities of Li-CuMH SSE films after soaking for various durations. As demonstrated in the **new Figure S15**, the ionic conductivity of Li-CuMH SSE with 0.5 h soaking time is only $2.5 \times 10^{-5} \text{ S cm}^{-1}$, and this value significantly increases to $1.1 \times 10^{-4} \text{ S cm}^{-1}$ for the Li-CuMH SSE with 24 h soaking time, revealing that the ionic conductivity of the Li-CuMH SSE increases with the increase of total amount of Li⁺ embedded. Given the above, the

correlation between the crystal structures and ionic conductivities of Li-CuMH SSEs with different Li^+ implantation times convincingly indicates that Li^+ ions can intercalate into the bulk crystal structure, ultimately resulting in varying degrees of ionic conductivities.

New Figure S14

Figure S14. a) XRD spectra of Li-CuMH SSE films after soaking in the non-aqueous electrolyte containing LiTFSI salts for different times and b) the corresponding enlarged spectra in the 2-theta range of 11.5° and 13° .

New Figure S15

Figure S15. a) Nyquist plots of Li-CuMH SSE films with different Li^+ implantation times and b) the corresponding ionic conductivities at room temperature.

According to the reviewer's constructive suggestions, we have revised the relevant parts on

page 6 of the revised manuscript, and on pages 14 and 15 of the supplementary information.

We copy the relevant text here for your check:

“**Fig. S13** further shows the X-ray diffraction (XRD) spectra of CuMH and Li-CuMH SSE films and the corresponding enlarged spectra in the 2-theta range of 11.5° and 13°. The characteristic peak corresponding to the (001) crystal plane changed significantly after the sufficient Li⁺ ion implantation, mainly manifested as the peak position shifting negatively from 12.27° to 12.21°, accompanied by a decrease in peak intensity. This clearly indicates that the crystal spacing indeed expands with the implantation of Li⁺ ions, and the diffraction intensity decreases accordingly. Moreover, the crystal structures of Li-CuMH SSE films after soaking in the non-aqueous electrolyte containing LiTFSI salts for different times were further investigated (**Fig. S14**). Apparently, the characteristic peaks located at around 12.3° continue to shift negatively, accompanied by a continuous decrease in peak intensity, indicating that Li⁺ ions are indeed constantly intercalated into crystal interlayers. This is highly consistent with what we found in **Fig. S13**.”

“Electrochemical impedance spectroscopy (EIS) measurements were further performed to estimate the change in ionic conductivities of Li-CuMH SSE films after soaking for various durations. As **Fig. S15** demonstrates, the Li-CuMH SSE films show significantly reduced polarization as the soaking time increased, and the ionic conductivity increases from 2.5×10^{-5} to 1.1×10^{-4} S cm⁻¹ when the soaking time increase from 0.5 to 24 h. Given the above, the correlation between the crystal structures and ionic conductivities of Li-CuMH SSEs with different Li⁺ implantation times convincingly indicates that Li⁺ ions can intercalate into the bulk crystal structure, ultimately resulting in varying degrees of ionic conductivities.”

Q2: A big problem with the proposed mode of conductivity is that the assumption is that LiTFSI dissociates and the lithiums move en masse into the crystal, and leave the TFSI anions behind. This is highly improbable. It would lead to an immense positive charge buildup in the interior of the crystal, and an immense negative charge buildup outside the crystal. It is difficult to separate even one ion pair. The notion that millimoles of ions would spontaneously separate just by "soaking" in a liquid electrolyte seems very unlikely without an immense driving force. There is no clear driving force for such a massive charge separation on bulk scale.

Our response: We are grateful for the very professional and insightful comment. In fact, we were also thinking about this critical question when preparing the first manuscript, and we fully agree with the reviewer that there is no clear driving force for such a massive charge separation on bulk scale. The reviewer's question prompted us to conduct more experiments and calculations to solve this problem. We first reinvestigated the Cu 2P XPS spectrum of the CuMH film (**original Figure S35** of the supplementary information) and found the presence of a considerable proportion of Cu⁺ cations. As we demonstrated in the experimental section, we chose Cu⁰ foil as copper precursor instead of Cu²⁺ salts to synthesize CuMH films (**Figure 1**). In our previous manuscript, we did not pay attention to the existence of Cu⁺, but this is precisely the key to Li⁺ embedding and maintaining charge balance. As a comparison, we synthesized CuMH film using Cu²⁺ salts as precursors, and studied the effect of different precursors on the Cu valence state in the products with the aid of XPS and Auger spectroscopy. As shown in **Figure R1a** (Note that **Figure RX** represents the figures used in the response letter and will not appear in the manuscript and supporting information), the binding energy (BE) peak at *ca.* 935 eV is assigned to Cu²⁺ species, accompanied by the characteristic Cu²⁺ shakeup satellite peaks (938-945 eV).^[1] The BE peak at ~933 eV clearly demonstrates the existence of Cu⁺ or Cu⁰ species.^[2, 3] Moreover, the peak located at ~570 eV in the Auger Cu LMM spectrum further confirms the presence of Cu⁺ (**Figure R1b**).^[4] In contrast, the CuMH film prepared from Cu(NO₃)₂ only shows the characteristic peak of Cu²⁺ species (**Figures R1c and R1d**). These results show that starting from Cu⁰ foil is a prerequisite for obtaining CuMH film containing Cu⁺ cations (see the equation below). Furthermore, the ratio of Cu⁺ and Cu²⁺ content in our CuMH is calculated by integrating the area of the corresponding peaks in **Figure R1b**, which shows that the proportion of Cu⁺ and Cu²⁺ in CuMH is 58.8% and 41.2%, respectively. After the sufficient Li⁺ implantation, the obtained Li-CuMH SSE shows the very close XPS and Auger Cu LMM spectra to CuMH, demonstrating the structural and compositional stability during the Li⁺ embedding process (**Figure R2**).

Figure R1. a) Cu 2p XPS and b) Auger Cu LMM spectra of CuMH film synthesized from different Cu precursors. c) Cu 2p and d) Cu LMM spectra of CuMH film prepared from MA and Cu⁰ foil. Cu 2p and Cu LMM spectra of CuMH film obtained from MA and Cu(NO₃)₂ salts.

Figure R2. a) Cu 2P XPS and b) Auger Cu LMM spectra of Li-CuMH SSE film after the sufficient Li^+ implantation.

According to the new experimental results, the crystal structure of Li-CuMH has been recalculated and shown in **Figure R3**. The appearance of Cu^+ ions indicates that some undissociated H^+ ions exist on the carboxylic acid that coordinates with Cu^+ ions, resulting in the presence of $(\text{HOOC}_2\text{H}_2\text{COO})^-$ in the structure of the CuMH crystal (**Figure R3a**). Furthermore, Li^+ ions undergo an ion-exchange reaction with the H^+ ions on the carboxylic acid to form charge-equilateral Li-CuMH model when the CuMH film is immersed in the organic liquid electrolyte (**Figures R3b-3c**). In this sense, the amount of Cu^+ ions in Li-CuMH is the same as the number of Li^+ ions. Therefore, there is no immense positive charge buildup in the interior of the CuMH crystal, which is due to the effective ion exchange between Li^+ from electrolyte and H^+ in CuMH.

Figure R3. a) Local structures of Cu^+ and Cu^{2+} coordinated with maleic acid in CuMH, respectively. b) The crystal structure of CuMH containing Cu^+ and Cu^{2+} along the c -axis. c) The crystal structure of Li-CuMH produced by the ion-exchange reaction between Li^+ ions with H^+ from CuMH.

According to the enlightening results from experiments and simulations, the relevant parts have been revised on pages 4 and 5 of the revised manuscript, and pages 10 and 11 of the supplementary information. We copy the relevant text here for your check:

“As a reference, CuMH prepared through traditional hydrothermal methods using $\text{Cu}(\text{NO}_3)_2$ as the precursor only exhibits irregular nanosheet structure (**Fig. S2**).”

“To further study the valence information of Cu cations in the CuMH crystal, XPS and Auger Cu LMM spectra of the CuMH sample were collected and analyzed (**Fig. S4**). The binding energy (BE) peak at ~ 935 eV is assigned to Cu^{2+} species, accompanied by the characteristic Cu^{2+} shakeup satellite peaks (938-945 eV)^[34]. The BE peak at ~ 933 eV clearly demonstrates the existence of Cu^+ or Cu^0 species (**Fig. S4a**)^[35, 36]. And the Auger Cu LMM spectrum further confirms the presence of Cu^+ at ~ 570 eV (**Fig. S4b**)^[37]. In contrast, the CuMH film prepared from $\text{Cu}(\text{NO}_3)_2$ only shows the characteristic peak of Cu^{2+} species (**Fig. S5**). These results manifest that starting from Cu^0 foil is a prerequisite for obtaining CuMH film containing Cu^+ cations. Furthermore, the ratio of Cu^+ and Cu^{2+} content in our CuMH is calculated by integrating the area of the corresponding peaks in **Fig. S4b**, which shows that the proportion of Cu^+ and Cu^{2+} in CuMH is 58.8% and 41.2%, respectively.”

“According to the valence information analysis, the crystal structure of Li-CuMH was further calculated and presented in **Fig. S6**. The Cu (II) ion possesses a square pyramidal coordination environment, and the apical oxygen atoms originate from the localized crystallized water molecules. The base atoms come from two oxygen atoms of a single maleate chelating ligand, and two oxygen atoms come from two additional maleate groups. The Cu (I) ion has a similar square pyramidal coordination environment as the Cu (II) ion, with the difference that one of the coordination oxygens is a hydroxyl oxygen, not a conjugated carbonyl oxygen. Each maleate group is bonded to three copper atoms and forms a polymeric monolayer. These polymeric monolayers are closely connected by strong hydrogen bonds between water molecules and the carbonyl oxygen atoms to form a periodic stacked structure (**Fig. S7**)^[38].”

New Figure S4

Figure S4. a) Cu 2p XPS and b) Cu Auger LMM spectra of CuMH film prepared from MA and Cu⁰ foil.

New Figure S5

Figure S5. a) Cu 2p XPS and b) Cu Auger LMM spectra of CuMH film prepared from MA and Cu(NO₃)₂ salts.

New Figure S6

Figure S6. Local structures of Cu^+ and Cu^{2+} coordinated with maleic acid in CuMH crystal, respectively.

New Figure S7

Figure S7. The crystal structure of CuMH containing Cu^+ and Cu^{2+} along the c -axis.

“In addition, the appearance of Cu^+ ions indicates that some undissociated H^+ ions exist on the carboxylic acid that coordinates with Cu^+ ions, resulting in the presence of $(\text{HOOC}_2\text{H}_2\text{COO})^-$ in the structure of the CuMH crystal. And this coordination structure is also found in other Cu-MOF materials containing carboxylic acid^[39, 40].”

“As demonstrated in **Fig. S10**, the Li^+ ion undergoes an ion-exchange reaction with the H^+ ions on the carboxylic acid to form charge-equilateral Li-CuMH model during the Li^+ impregnation process.”

New Figure S10

Figure S10. The crystal structure of Li-CuMH produced by the ion-exchange reaction between Li^+ ions with H^+ from CuMH.

Q3: *The SAXS pattern shows no change after Li loading. This strongly suggests there is no Li^+ intercalation. The inclusion of Li into the lattice should swell the crystal (we all know that cathodes and anodes swell when they are loaded with Li^+ . Why wouldn't this material?), and alter the XRD pattern, and. The SAXS peaks do not change upon lithiation, suggesting there is no change in the unit cell, and no lithium uptake.*

Our Response: We greatly appreciate the reviewer for the very professional and insightful comments. As we answered to Q1, we have restudied the crystal structures of as-prepared CuMH and Li-CuMH films and confirmed the variation in the XRD patterns (as shown in the new **Fig. S13**). That is, Li^+ ions can indeed intercalate and fill the interlayers. In this sense, the SAXS spectra of CuMH and Li-CuMH should exhibit significant differences. Therefore, we have re-determined the SAXS spectrum of the Li-CuMH and strictly controlled the testing conditions of the sample. We found that sample preparation in the air environment can easily lead to sample deterioration, which is the main reason why we were unable to detect the difference in the SAXS spectra of CuMH and Li-CuMH in our last draft. As we all know, many SSEs are unstable in the ambient air, such as H^+/Li^+ exchange for garnet-based SSEs, H_2S generation for sulfide-based SSEs and water absorption for polymeric solid electrolyte.^[5]

To avoid the influence of the air environment, we instead prepared CuMH and Li-CuMH samples in the glove box ($\text{H}_2\text{O} < 0.01$ ppm, $\text{O}_2 < 0.01$ ppm) and stored them in an airtight box, ultimately transferred them quickly to the SAXS chamber. The newly-tested SAXS patterns

clearly show the difference between CuMH and Li-CuMH samples (new **Fig. 2a**, new **Fig. 2b** and new **Fig. S12**). As demonstrated in **Figs. 2a, 2b and S12**, the CuMH sample exhibits two strongest peaks at the angstrom scale of $d_1 \approx 7.20 \text{ \AA}$ and $d_2 \approx 4.26 \text{ \AA}$, while the Li-CuMH sample exhibits two strongest peaks at the angstrom scale of $d_1 \approx 7.25 \text{ \AA}$ and $d_2 \approx 4.30 \text{ \AA}$, strongly proving the swelling effect of the CuMH crystal after the sufficient Li^+ implantation.

According to the enlightening suggestion, the relevant parts have been revised on pages 6 and 7 of the revised manuscript, and page 13 of the supplementary information. **We copy the relevant text here for your check:**

New Figure 2a

New Figure 2b

Figure 2. 2D SAXS patterns of a) CuMH and b) Li-CuMH samples.

New Figure S12

Figure S12. a) The SAXS curves of the CuMH and Li-CuMH, and the corresponding zoom-in curves in the q range of b) 0.8-0.95 \AA^{-1} and c) 1.35-1.6 \AA^{-1} , respectively. The corrected scattering intensity was plotted relative to the scattering vector q .

“As demonstrated in **Figs. 2a, 2b** and **S12**, the CuMH sample exhibits two strongest peaks at the angstrom scale of $d_1 \approx 7.20 \text{ \AA}$ and $d_2 \approx 4.26 \text{ \AA}$, while the Li-CuMH sample presents two strongest peaks at the angstrom scale of $d_1 \approx 7.25 \text{ \AA}$ and $d_2 \approx 4.30 \text{ \AA}$. Apparently, after soaking in the organic liquid electrolyte, the angstrom-scale channels of CuMH accordingly expand and the corresponding peak intensity weakens, convincingly confirming the effective embedding of Li^+ ions into CuMH crystals.”

Q4: The authors note a difference in the SAXS peaks, but their observation is problematic:

“Note that the weakening of the intensity of all peaks represents a decrease in the electron density of Li-CuMH units, which could likely be attributed to the possible interaction between Cu (II) and TFSI⁻ anions^[30].”

This is not really how one would expect an XRD to change in response to an alteration of composition. Changes in atomic structure should change the peak positions of the XRD pattern. Even in the unlikely event the unit cell is not perturbed, but new atoms are inserted, a change in atomic composition will increase the intensity of some peaks (that have stronger or more scatterers on those hkl planes) and decrease the intensity of other peaks. The authors argue

that a uniform decrease in all peak intensities could be meaningful, when the only reasonable interpretation for an identical but weaker pattern, is that less sample was used. ?

Our Response: We first thank the reviewer for the very professional question. As we answered to Q1 and Q3, we have restudied the crystal structures of as-prepared CuMH and Li-CuMH films and confirmed the variation in the XRD and SAXS patterns. For the supplementary XRD tests, the CuMH and Li-CuMH samples were sealed by Kapton tape to avoid the sample deterioration in the atmospheric environment. The newly-measured XRD patterns reveal that the characteristic peak of the Li-CuMH film negatively shifts from 12.27° to 12.21° , accompanied by a decrease in peak intensity after the Li^+ intercalation (new **Fig. S13**). Note that no new impurity peaks appeared after the samples were encapsulated for protection. Moreover, the newly-tested SAXS patterns of CuMH and Li-CuMH films came to the same conclusion (as we discussed in the new **Figs. 2a, 2b** and **S12**). More importantly, the ion conductivities of Li-CuMH SSEs have been confirmed to increase with the increase of total amount of Li^+ embedded (new **Figure S15**). Apparently, all above results convincingly indicate that Li^+ ions can intercalate into the bulk crystal structure, and different soaking times will ultimately result in varying degrees of ionic conductivities.

New Figure S13

Figure S13. a) XRD spectra of CuMH film and Li-CuMH SSE film and b) the corresponding enlarged spectra of 2 theta in the range of 11.5° and 13° .

New Figure S15

Figure S15. a) Nyquist plots of Li-CuMH SSE films with different Li⁺ implantation times and b) the corresponding ionic conductivities at room temperature.

Q5: The authors base much of the arguments and characterization on a single crystal structure, but this data is not provided. The authors must include a CIF so that reviewers can evaluate the single-crystal structure.

Our Response: We thank the reviewer for the very professional suggestion and fully agree that the related CIF files should be provided. The X-ray diffraction of the previous literature^[6] showed that the unit cell parameters of Cu^{II}MH were $a = 7.701 \text{ \AA}$, $b = 5.262 \text{ \AA}$, $c = 7.743 \text{ \AA}$, $\alpha = \gamma = 90^\circ$, $\beta = 111.833^\circ$, indicating a monoclinic system with the $P21$ space group. The crystal structures of Cu^{I/II}MH and Li-Cu^{I/II}MH were calculated based on the $2 \times 3 \times 1$ supercell structure of Cu^{II}MH. The CIF of Cu^{I/II}MH and Li-Cu^{I/II}MH were displayed in supporting information. And the CCDC (Cambridge Crystallographic Data Centre) number for the single crystal structure of CuMH is 1133197.^[6] We also provided three CIF files associated with Cu^{II}MH, Cu^{I/II}MH and Li-Cu^{I/II}MH when uploading the revised manuscript.

Q6: In Figure S8, the authors argue that the appearance of a few new peaks suggests Li intercalation, but this is also unsubstantiated. If a crystal structure changes, one does not expect to get new XRD peaks in addition to the old ones. The whole pattern must change. A PXRD is not like an NMR where you get new signals for new atoms. It is a Fourier transform

of the electron density map. Everything in the XRD pattern changes drastically if you perturb the periodic lattice.

Our Response: We first thank the reviewer for the very professional question. As we answered to Q4, we have restudied the crystal structures of as-prepared CuMH and Li-CuMH films and confirmed the variation in the XRD and SAXS patterns. For the supplementary XRD tests, the CuMH and Li-CuMH samples were sealed by Kapton tape to avoid the sample deterioration in the atmospheric environment. The newly-measured XRD patterns reveal that the characteristic peak of the Li-CuMH film negatively shifts from 12.27° to 12.21°, accompanied by a decrease in peak intensity after the Li⁺ intercalation (new **Figure S13**). More importantly, no new impurity peaks appeared after the samples were encapsulated for protection. Consistent with the reviewer's judgment, Li-CuMH and CuMH do have basically the same diffraction peaks. Based on the newly measured XRD results, we have removed our previously inaccurate expression on page 6 of the revised manuscript.

Q7: The authors also use NMR broadening arguments to support the intercalation of Li but these are not convincing either. These results would be expected whether the Li was in the solid grains, or the boundaries between.

Our Response: We thank the reviewer for the valuable suggestion and apologize for the inaccurate statements. In this work, the ¹Li NMR spectrum serves as the supporting detail to explain the enhanced ionic conductivity of Li-CuMH SSE.^[7] According to the reviewer's suggestion, we have revised the expression on the page 13 of the revised manuscript and copy the relevant text for your check:

“Moreover, the ⁷Li resonance of the pristine Li-CuMH is 1.60 ppm and shifts upfield after cycling, representing a change in the polarization environment of the Li⁺ ions in the Li-CuMH. This change usually indicates a strong interaction between Li⁺ ions and polar groups in the Li-CuMH SSE, which enhances the ionic conductivity^{[55, 56].”}

*Q8: The authors offer ESP mapping as a surrogate evidence for lithium intercalation, but this is not useful. Of course the oxygenic ligand sites are more negative and *would* have a higher affinity for Li⁺ than the uncharged atoms, but this is not evidence of LiTFSI mass dissociation*

and intercalation into a crystal without lattice disruption.

Our Response: We thank the reviewer for the very rigorous question and we agree with the reviewer's point. As previously mentioned, the intercalation of Li^+ ions in the CuMH crystal has been confirmed by the XRD and SAXS characterizations. The ESP mappings of CuMH (new **Figure 2c** and **Figure S17**) indicate that carboxylic acid oxygen in CuMH can provide sites for Li^+ ions intercalation, which is consistent with the occupied sites of Li^+ ions in the Li-CuMH crystal structure. As shown in **Figure 2g** of the revised manuscript, Li^+ ions occupy the vacancies around the carboxylic acid oxygen and migrate along the *b*-axis direction of the extension of the carboxylic acid. According to the reviewer's suggestion, we have revised the expression on the page 7 of the revised manuscript. We copy the relevant text for your check: "As manifested in **Figs. 2c** and **S17**, the electrostatic potential (ESP) mappings indicate the high local electronegativity of oxygen atoms from the maleate unit and structural H_2O in the CuMH crystal."

New Figure 2c

Figure 2. c) Electrostatic potential (ESP) distribution of the CuMH unit.

New Figure S17

Figure S17. Electrostatic potential (ESP) distribution of maleate unit and structural water in Li-CuMH.

Q9: The migration paths shown in Figure 2g offer a testable hypothesis for the expected XRD changes in response to Li intercalation. If lithiums do indeed slip into this region without expanding the crystal, you are inserting ions (and hence electron density) into the 200 plane. If this hypothesis is correct, this increase in electron density should result in an increase in the intensity of the 200 reflection due to the additional scattering from the new Li ions, and a decrease in the intensity of 100 because the new Li ions are out of phase with the Cu atoms on the 100 plane, and give some destructive interference. But the SAXS seems to suggest there is no change in the intensities (or positions) of the XRD peaks at all.

Our response: We thank the reviewer for the very professional suggestion. As we answered to Q1, Q2 and Q3, we have remeasured the XRD and SAXS of CuMH and Li-CuMH and confirmed the structural variation after the Li⁺ implantation. More importantly, we experimentally revealed the existence of Cu⁺ in the CuMH with the aid of XPS and Auger spectra. The appearance of Cu⁺ ions indicates that some undissociated H⁺ ions exist on the carboxylic acid that coordinates with Cu⁺ ions, resulting in the presence of (HOCC₂H₂COO)⁻ in the structure of the CuMH crystal (new **Figs. S6, S7**). Furthermore, Li⁺ ions undergo an ion-exchange reaction with the H⁺ ions on the carboxylic acid to form charge-equilateral Li-CuMH model when the CuMH film is immersed in the organic liquid electrolyte (new **Fig. S10**). On

this basis, we have restarted the simulations for the Li^+ ion migration pathways based on the new structures of Li-CuMH. The relevant results are shown in the new **Figs. 2d-g, S18, S19** and **S20**. According to the new calculations, we have revised the relevant parts on pages 7 and 8 of the revised manuscript. We copy the relevant text here for your check:

“The direction of the [Li1-Li2-Li3-Li4] chain along the *b*-axis was found to be the most favorable 1D migration path for Li^+ ions (**Fig. S18**), with a minimum migration energy barrier of 0.619 eV. Li^+ ions were observed to migrate along the [Li1-Li2-Li3-Li4] chain and interconnect with the [Li1-Li1] chain to form a 2D migration path in the *bc* plane (**Fig. S19**), with a corresponding migration energy barrier of 0.655 eV. In contrast, the 2D path connects the remaining active sites into a 3D path (**Fig. S20**), but the migration energy barrier is high up to 1.171 eV.”

New Figures 2d-2g

Figure 2. Structure and Li^+ ion migration pathways. Energy profiles of d) 1D, e) 2D and f) 3D possible Li^+ ion migration pathways in Li-CuMH. g) The most favorable Li^+ migration pathway along the [010] direction of the *b*-axis, viewed as the light purple isosurface of constant $E_{\text{BVSE}(\text{Li})}$ superimposed on the crystal structure.

New Figure S18

Figure S18. 1D Li^+ ion migration path in the direction of the $[010]$ chain along the b -axis (red dotted line).

New Figure S19

Figure S19. 2D Li^+ ion migration path in the bc plane.

New Figure S20

Figure S20. 3D Li⁺ ion migration path in the Li-CuMH.

Q10: The simulated models of ion diffusion represent an opportunity to test the assumption that the crystal lattice does not swell upon lithium intercalation. In the models of lithium ion channels, is the simulated model allowed to swell to accommodate the Li ions? Does the lithium actually fit here without expanding the lattice? What are the O---Li distances? Are they realistic? Does this AIMD simulation predict a swelling of the crystal lattice? If so, the authors should be able to approximate the new cell parameters that would be expected for Li intercalation from their simulation.

Our response: We appreciate the reviewer's rigorous comments and fully agree with the reviewer that the crystal lattice would swell upon Li⁺ ions intercalation. As we answered to Q1, Q2 and Q3, we have remeasured the XRD and SAXS of CuMH and Li-CuMH and confirmed the structural variation (*i.e.*, expansion) after the Li⁺ implantation. In our calculations, the lattice size of Li-CuMH was kept constant when simulating the migration path of Li⁺ ions in Li-CuMH based on AIMD, but lattice variations were taken into account in calculating the structural evolution of the Li⁺ ions embedded in CuMH, and the crystal parameters obtained are shown in **Table R1**. As can be seen from **Table R1**, the lattice undergoes significant

expansion along the b -axis in the direction of Li^+ ion intercalation and migration. However, compared with the long-range hydrogen bonding interaction in $\text{Cu}^{\text{I/II}}\text{MH}$, the insertion of Li-ions forms stronger Li-O bonds in $\text{Li-Cu}^{\text{I/II}}\text{MH}$, leading to a decrease in the lattice volume of $\text{Li-Cu}^{\text{I/II}}\text{MH}$. In the structure of Li-CuMH , the Li^+ ion is coordinated to three oxygen from carboxylic acids and one oxygen from water (**Figure R4a**), where the bond lengths of Li-O are in the range from 1.915 to 2.168 Å (**Figure R4b**), which are within the normal range of Li-O ion bond.^[8, 9]

Table R1. The lattice parameters of $\text{Cu}^{\text{I}}\text{MH}$ ($1\times 3\times 2$), $\text{Cu}^{\text{I/II}}\text{MH}$ and $\text{Li-Cu}^{\text{I/II}}\text{MH}$.

	$a/\text{Å}$	$b/\text{Å}$	$c/\text{Å}$	$V/\text{Å}^3$	$\alpha/^\circ$	$\beta/^\circ$	$\gamma/^\circ$
$\text{Cu}^{\text{I/II}}\text{MH}$	15.402	15.437	7.855	1714.273	95.660	112.541	90.664
$\text{Li-Cu}^{\text{I/II}}\text{MH}$	15.032	16.152	7.539	1699.915	90.149	111.773	90.751

Figure R4. a) The coordination structure of Li atom and b) bond length of Li-O in Li-CuMH .

Minor problems/suggestions

Q11: The authors refer to the local electronegativity of COO^- and H_2O , which is improper. Groups/molecules don't have electronegativities; atoms do. The authors mean to say that the ligands are negatively charged as expected due to the electronegativity of the oxygen atoms.

Our response: We thank the reviewer for the very professional comments. According to the reviewer's suggestion, we have made minor revisions on page 7 of the manuscript and copy the relevant text here for your check:

“As manifested in **Figs. 2c** and **S17**, the electrostatic potential (ESP) mappings indicate the high local electronegativity of oxygen atoms from the maleate unit and structural H₂O in the CuMH crystal.”

Q12: Li/Li symmetric cell is not a "battery" strictly speaking, because it cannot spontaneously charge and discharge. Please refer to the Li/SSE/Li system as a "symmetric cell" instead of a battery.

Our response: We appreciate the reviewer’s rigorous comments. According to the reviewer’s suggestion, the expressions of “Li/Li symmetric batteries”, “Li/Li-CuMH SSE/Li symmetric battery”, “Li/Li-CuMH SSE/Li symmetric batteries”, and “Li/pure PEO/Li symmetric battery” are corrected to “Li/Li symmetric cells”, “Li/Li-CuMH SSE/Li symmetric cell”, “Li/Li-CuMH SSE/Li symmetric cells”, and “Li/pure PEO/Li symmetric cell”, respectively, which are marked in red in the revised manuscript.

Q13: Cycling with increasing voltage until failure (Figure 4a) strikes me as a strange approach to test the cycling limits of the electrolyte. Dendritic failure comes from either high current or high cycle number, and this experiment is changing them both at once, and then attributing the failure to the voltage only, and not the cycle number. Is this a common practice? I'm unfamiliar with it. If one cycles the symmetric cell at 0.8 V does it really hold up well?

Our response: We thank the reviewer for the meticulous comments, and we apologize for causing this misunderstanding. Actually, the critical current density (CCD) is the maximum endurable current density that lithium-ion batteries can withstand when cycling without cell failure, which has been developed for the past 40 years.^[10] Therefore, CCD is a key parameter that can help to determine the rate-determining steps of Li reaction kinetics in solid-state batteries. The CCD is measured by stepwise increasing currents with different protocols, such as time control, capacity control, endurable time control and long-term cycling.^[11] The CCD shown in **Figure 4a** of the manuscript is examined by the protocol of endurable time control, which can simultaneously evaluate the effect of current density and cycle number on the battery failure mechanism. In our manuscript, the long-term cycling stability tests were also conducted at various current densities and areal capacities to evaluate the accumulation effects of interfacial side reactions^[12] and presented in **Figs. 4c-d** and **S29a**.

Q14: The authors conclude that the electrolyte has good interfacial properties with the electrodes, but I am concerned its actually a slurry with residual liquid electrolyte. The authors prepared the electrolytes by soaking in liquid electrolyte, rinsing with DMF, and then padding them dry, and evacuating. But these solvents are very low-vapor pressure.

Our response: We first thank the reviewer for the professional comment. Notably, the solvent used in this work was **DME** instead of DMF. The boiling point of DME is 82 °C, while the boiling point of DME is as high as 153 °C. On page 3 of the supplementary information, the electrolytes were prepared by soaking in liquid electrolyte, rinsing with DME solvent, and then vacuum drying at 60 °C. We copy the original text about the electrolyte preparation for your check:

“Preparation of the Li-CuMH SSE

The as-prepared CuMH films were soaked into the 1M LiTFSI -1, 3-dioxolane /1, 2-dimethoxyethane (DOL/DME, 1:1, v/v) electrolyte at 60 °C for at least 24 h to ensure that Li⁺ ions fully penetrate into the regular channels of CuMH. Then, the soaked membranes were removed from the liquid electrolyte, cooled to room temperature and washed with the DME solvent to remove the adsorbed LiTFSI salts. These procedures were repeated at least three times to achieve full implantation of Li⁺ ions. After sufficient impregnation, the residual solvent on the CuMH film was removed by pressing and wiping with the dust-free cloth. The obtained Li-CuMH SSE films were then dried at 60 °C for 2 h and then vacuum dried for 2 h at the same temperature. The above two steps were repeated until all solvents and adsorbed water were thoroughly removed from the Li-CuMH SSE film (except for structural water). Note that all obtained Li-CuMH SSEs were aged for at least 3 h prior to battery assembly.”

Q15: I would suspect this is a liquid electrolyte percolating between the particles. The particles are probably not directly contacting the electrode, which explains why the SEI analysis turns up only components of the anion (F, S, C, O), and not the discriminating component of the electrolyte: copper. An SEI formed from the solid electrolyte should contain copper atoms.

Our response: Thank you very much for the very professional question. As we answered to Q1, Q2 and Q3, we have remeasured the XRD and SAXS of CuMH and Li-CuMH and confirmed that Li⁺ ions can effectively intercalate into the crystal interlayers. As we discussed

in our manuscript, both CuMH and Li-CuMH films are chemically and electrochemically stable in a wide voltage window. In this sense, we can assume that CuMH will not participate in any reaction, and therefore no copper-containing components will be found in the SEI layer. The components of anions (F, S, C, O) in SEI are likely derived from the decomposition of residual LiTFSI ($C_2F_6LiNO_4S_2$) salts during the preparation of Li-CuMH SSEs.

In summary, the authors do appear to have discovered something interesting. The electrochemical data shows that this electrolyte works well and is very promising. However, for Nature Commun., a much more well developed characterization of the active material, and better supported claims are needed. If the authors can demonstrate convincingly with data (not just computation) that there is lithium in the crystal lattice, this paper could be reconsidered. However, if this effect is a lithium intercalation along a percolating network (e.g., along the surface of the crystals) as I suspect, this could still be a useful technology, but is less novel in that it is similar to the addition of nanophases to polymer electrolytes, which generates percolating surface networks of mobile ions, and in this case, should be submitted elsewhere once the mechanism and mode of ion transport is better supported.

Our response: We once again appreciate the reviewer for the very professional and insightful comments. All above wise questions prompted us to rethink the working mechanism of the Li-CuMH SSE from a microscopic point of view. We accordingly supplemented a large number of related experiments to convincingly prove that Li^+ ions can indeed intercalate and fill the interlayers. We also found the presence of a considerable proportion of Cu^+ cations in the CuMH films based on the XPS and Auger analysis, and this is precisely the key to Li^+ embedding and maintaining charge balance. The appearance of Cu^+ ions indicates that some undissociated H^+ ions exist on the carboxylic acid that coordinates with Cu^+ ions, resulting in the presence of $(HOCC_2H_2COO)^-$. Furthermore, Li^+ ions undergo an ion-exchange reaction with the H^+ ions on the carboxylic acid to form charge-equilateral Li-CuMH model when the CuMH film is immersed in the organic liquid electrolyte. On this basis, we have also restarted the calculations for the Li^+ ion migration pathways based on the new structures of Li-CuMH and reasonably interpreted the migration mechanism of Li^+ ions within the Li-CuMH bulk. More importantly, the professional questions raised by the reviewers regarding XRD and

SAXS spectra also prompted us to exercise stricter control over the test conditions, avoiding sample deterioration and impurity peaks caused by the previous environmental impacts, and enabling us to obtain more convincing data.

In summary, the professional comments of the reviewers have given us a deeper understanding of the working mechanism of this new solid-state electrolyte, and we have answered all questions through supplementing the experiments and calculations.

Reviewer #2 (Remarks to the Author): =====

In the manuscript titled “Self-assembled hydrated copper coordination 1 compounds as ionic-conductors for room temperature solid-state batteries”; Zhang and his colleagues have successfully integrated the characteristics of inorganic solid-state electrolytes, specifically the Li diffusion channel, with the coupling dissociation phenomenon of polar organic solid-state polymer electrolytes. This integration has led to enhanced Li transportation in a 2D lithiated Cu-maleate coordination compound, which serves as an organic-inorganic hybrid. The compound exhibits a remarkable breakthrough in Li conductivity, as demonstrated by its exceptional performance in a Li/Li-CuMH SSE/Li symmetric cell and in an LFP/Li-CuMH/Li full cell.

The manuscript presents a meticulous investigation of the plausible mechanism, supported by both computational modeling and experimental evidence. The methodical use of graphical and pictorial representations effectively conveys the findings. The manuscript is well-organized, with a wealth of data and a profound understanding of the subject matter, making it highly suitable for acceptance in Nature Communications after revision.

Our response: We are grateful to the reviewer for the positive and professional comments. All requested revisions based on the constructive comments from the reviewer have been fully considered and thoroughly addressed, as shown below.

Furthermore, the recommended suggestions to refine and enhance the transparency of the manuscript will likely improve its suitability for publication.

Some scientific suggestions:

Q1: *In the manuscript, the authors initially mention that CuMH exhibits a 2D nanoflake structure, as confirmed by SEM. Considering the lithiation process involved in the formation of Li-CuMH SSE, it is essential to determine whether the nanoflake morphology is retained. If so, it is crucial to clearly explain the advantages of such a morphology in facilitating the conduction of Li-ions in bulk Li-CuMH SSE.*

Our response: We first thank the reviewer for the very insightful comment and fully agree that the morphology of the Li-CuMH should be taken to confirm the preservation of the 2D

nanosheet structure. In this work, the Li-CuMH powder was fabricated by soaking in the liquid organic electrolyte, rinsing with DME solvent, and vacuum drying. As shown in **Figure R5**, the 2D nanoflake morphology of the Li-CuMH sample after the Li^+ implantation was well preserved. Both CuMH and Li-CuMH samples exhibit regular disc-shaped structures with an average diameter of 2-3 μm , and the cross-sectional SEM images further reveal that these micro-disks are tightly stacked by layered nanosheets (**Figures R5b** and **R5d**). From a more microscopic perspective, these tightly stacked nanosheets comprise a large number of 1D Li^+ ion migration paths (**Figs. 2g** and **S18**), thus exhibiting high ionic conductivity at room temperature.

Figure R5. a) Top view and b) side view SEM images of 2D CuMH nanoflakes. c) Top view and d) side view SEM images of 2D Li-CuMH nanoflakes after the Li^+ implantation.

Q2: In order to determine the state of the crystallized water between the layers of CuMH during the lithiation process for Li-CuMH SSE production, it is recommended to perform TGA-DTA analysis and ^1H -Proton NMR verification.

Our response: We first thank the reviewer for the very professional comment. According to the reviewer's suggestion, the state of structural water in CuMH powder, CuMH film and Li-CuMH film were analyzed by the TGA curves and shown in **Figure S16**. All samples remained stable until the temperature reaches 100 °C, and then gradually dehydrated in the temperature range of 100-160 °C^[13]. This directly demonstrates that the water in the CuMH and Li-CuMH samples is indeed crystallized, rather than free water. As manifested in **Figure R6**, the peak of ¹H ssMAS NMR spectra located at ~4.9 ppm also corresponds to the structural water.^[14] After the Li⁺ intercalation, the positive chemical shift in the ¹H signal of the CuMH sample can be attributed to the decrease in electron density caused by the interaction between H₂O and Li⁺ ion. Apparently, the NMR results also support the presence and stability of structural water in both the CuMH and Li-CuMH.

Figure S16. TGA curves of CuMH powder, CuMH film and Li-CuMH film. All samples remained stable until the temperature reaches 100 °C, and then gradually dehydrated in the temperature range of 100-160 °C. This directly demonstrates that the water in the CuMH and Li-CuMH samples is indeed crystallized, rather than free water.

Figure R6. ^1H ssMAS NMR spectra of the CuMH and Li-CuMH films.

Concern related to the contribution from proton conductivity from the structural water

(i) *When considering the presence of water in the Li-CuMH SSE, as indicated by the single crystal structure of CuMH, it is important for the authors to address how they confirm that H_2O does not play a major role in proton conduction. There is a possibility of proton conduction being predominant, as protons are more mobile than Li ions. Alternatively, a dual effect of proton conduction and Li conduction could occur. This greatly depends on the Li/ H_2O ratio within the overall solid electrolyte. To provide clarification, it is essential for the authors to provide the specific Li/ H_2O ratio in the Li-CuMH SSE during the study.*

(ii) *The use of MSD (Mean Square Displacement) is an excellent technique to assess the migration rates of different constituents within a structure. While the authors calculated the MSD for COO^- , $\text{HC}=\text{CH}$, and Cu(II) in Li-CuMH, it is important for them to also include the calculation for structural H_2O . This analysis would provide further insight into whether the observed partial contribution is attributed to proton conductivity or other factors."*

(iii) (a) *It would be beneficial for the authors to calculate the same ESP (Electrostatic Potential)*

mapping on the exact building unit of CuMH. This approach would enhance the prominence of the analysis and provide a better understanding of the basic character of the maleate unit and the structural H₂O. (b) In addition to the computational investigation, it is recommended that the authors complement their research with proton NMR investigation to assess the state of the structural H₂O during the Li transportation process. Performing the NMR analysis either *ex situ* or *in situ* would provide valuable insights. (c) To confirm the absence of a significant role of proton conductivity, it is crucial for the authors to conduct a control experiment using Li-CuMH and CuMH separately. The CuMH sample should not exhibit considerable ionic conductivity in the absence of Li.

Our response: We first thank the reviewer for the very rigorous comments. In this work, the structural water cannot offer proton conductivity due to its localization at specific positions in the CuMH crystal (**Figs. 2g** and **S18**). A direct and convincing evidence is the wide electrochemical window of the Li-CuMH SSE (**Figure 3e**), which is sufficient to indicate that there is no free water in the crystal, and therefore no proton conductivity. According to the reviewer's suggestion, the Li/H₂O ratio in Li-CuMH SSE is examined by inductively coupled plasma-optical emission spectrometry (ICP-OES, Agilent ICP-OES 730) and TGA technologies. As shown in **Table R2**, the weight contents of Cu and Li elements are 12.6 wt.% and 1.03 wt.%, respectively, and the corresponding molar ratio of Cu to Li is 1:0.75 ($M_{\text{Cu}}=63.5 \text{ g mol}^{-1}$, $M_{\text{Li}}=7 \text{ g mol}^{-1}$). Based on the TGA results, the molar ratio of Cu to H₂O is 1:1, thus the molar ratio of H₂O to Li is 1:0.75. Notably, the XRD spectra of the Li-CuMH SSE indicate that H₂O is still in coordination with Cu cations after the sufficient Li⁺ embedding (**Figures S13 and S14**).

Table R2. ICP-OES result of the Li-CuMH SSE.

Samples	Element	Element content in the	Element content in sample
		solution (mg L ⁻¹)	(wt.%)
Li-CuMH	Cu	0.5946	12.60
	Li	0.0410	1.03

The MSD technique is employed to assess the migration rates of H₂O. As shown in new **Figure S21**, the maximum diffusion coefficients of H⁺ ion and O²⁻ ion in the structural H₂O are 9.28×10^{-6} and 7.99×10^{-6} cm² s⁻¹ at 750 K, respectively, while that of Li⁺ ions (D_{Li^+}) are up to 1.53×10^{-5} cm² s⁻¹ at 750 K, respectively. Further, radial distribution function (RDF) analysis was performed to analyze the radial distribution of hydrogen bonds. The RDF represents the probability density of hydrogen bond (O-H), and the higher the value of a point, the higher the number of hydrogen bonds at that point. As demonstrated in new **Figure 3d**, the O-H bonds of 1.05 Å in H₂O from the Li-CuMH structure have the highest aggregation at different temperatures, and the number of hydrogen bonds is approximately the same, indicating that the structural H₂O in the crystal framework is not decomposed into hydrogen protons. This further indicates that the structural water does not offer proton conductivity in the Li-CuMH SSE.

New Figure S21

Figure S21. Mean square displacement (MSD) plots of a) Li, b) H, c) O d) C and e) Cu in Li-CuMH at 350, 450, 550, 650 and 750 K, respectively. f) Arrhenius plots plotted according to (a-e).

New Figure 3d

Figure 3. d) Radial distribution function of O-H in the structural H₂O using the trajectory obtained by MD calculation.

The ESP mapping on the unit of CuMH is also investigated for a better understanding of the basic character of the maleate unit and the structural H₂O. **New Figure 2c** clearly shows that the positions with the smallest ESP values (red parts) are all close to the carbonyl oxygen on the maleic acid, corresponding to the possible insertion point of the lithium ion. Apparently, after the coordination of Cu with H₂O, the ESP values around the oxygen atom in H₂O become relatively positive (blue parts), indicating that Cu atom can covalently interact with the oxygen atom in H₂O.

To study the state of structural H₂O in Li-CuMH SSE during the Li⁺ transportation process, the ¹H ssMAS NMR spectra of Li-CuMH before and after cycling were recorded and compared (**Figure R7**). For the cycled Li-CuMH SSE, the ¹H signal remains almost unchanged compared with the pristine Li-CuMH SSE, indirectly indicating that hydrogen in the structural H₂O does not participate in the ionic conduction, *i.e.*, no proton conductivity.

Moreover, the ionic conductivity of CuMH film without soaking in the liquid organic electrolyte was measured by the EIS test (**Figure R8**). The Nyquist plot of the pristine CuMH SSE presents an ultrahigh bulk impedance (~5000 ohm), corresponding to an extremely low ionic conductivity of $7 \times 10^{-7} \text{ S cm}^{-1}$. This further strongly confirms that the ionic conductivity

of the Li-CuMH is almost contributed by the Li^+ migration, rather than the proton conductivity.^[15]

New Figure 2c

Figure 2. c) Electrostatic potential (ESP) distribution of the CuMH unit.

Figure R7. ^1H ssMAS NMR spectra of the Li-CuMH SSE before and after cycling.

Figure R8. a) Nyquist plot of the CuMH SSE at room temperature. b) Bar chart of room-temperature ionic conductivities of CuMH and Li-CuMH SSEs, respectively.

According to the reviewer’s instructive suggestion, we have revised the relevant part on pages 7-10 of the revised manuscript, and copy the relevant text here for your check:

“Mean square displacement (MSD) plots from 350 to 750 K are shown in **Figs. S21a-e**, including Li, H, O, C and Cu. The MSD plots indicate that Li^+ ions exhibit the fastest migration coefficient of $1.53 \times 10^{-5} \text{ cm}^2 \text{ s}^{-1}$, while H_2O , COO^- , $\text{HC}=\text{CH}$, and Cu (II) in the Li-CuMH backbone move much more slowly. Moreover, the activation energy (E_a) of the Li-CuMH is calculated to be 0.43 eV (**Fig. S21f**).”

“As manifested in **Figs. 2c** and **S17**, the electrostatic potential (ESP) mappings indicate the high local electronegativity of oxygen atoms from the maleate unit and structural H_2O in the CuMH crystal.”

“Radial distribution function (RDF) analysis was performed to analyze the radial distribution of hydrogen bonds (**Fig. 3d**), where the O-H bonds of 1.05 Å in H_2O from the Li-CuMH structure show the highest aggregation at different temperatures and the values of hydrogen bonds are approximately the same, convincingly indicating that the structural H_2O is highly stable in the framework and does not offer proton conductivity.”

Q3: The correlation between high voltage resistance and the low HOMO energy level of Li-CuMH should be compared with the theoretical oxidation potential (vs. Li/Li⁺) value derived from the HOMO energy level. Without a specific theoretically calculated oxidation value, it remains a gross approximation.

Our response: Thank you very much for the professional advice. Due to the complexity of the oxidative decomposition of Li-CuMH, it is difficult to accurately obtain its oxidation potential based on theoretical calculations. Therefore, to evaluate the redox properties of Li-CuMH, the energy levels of highest occupied molecular orbital (HOMO) were calculated by the density functional theory (DFT). Meanwhile, the value of HOMO has been used to reflect the oxidation resistance of the organic-based solid-state electrolyte, such as C-SPE^[16], Q-COF^[17] and P-PLL-CPEs^[18]. The calculated HOMO values of CuMA and Li-CuMA are -7.63 and -7.02 eV, respectively, which are in agreement with the experimental conclusion that Li-CuMA has a high oxidation potential (**Figure S24**).

As suggested by the reviewer, the relevant part has been revised on page 9 of the revised manuscript. We copy the relevant text here for your check:

New Figure S24

Figure S24. HOMO energies of the CuMH and Li-CuMH.

“As manifested in **Fig. S24**, the HOMO energies of CuMH and Li-CuMH are -7.63 and -7.02

eV, respectively. The high antioxidation ability of the Li-CuMH is mainly attributed to the strong electron-withdrawing effect of the oxygen-containing groups. These results indicate that the Li-CuMH SSEs possess excellent compatibility with Li metal and most high-voltage cathode materials, owing to their high electrochemical stability and wide working potential window.”

Q4: *It would be helpful if the authors could explain how they calculated the transport number of lithium by combining the chronoamperometry and EIS techniques.*

Our response: We thank for the reviewer for the constructive suggestion and apologize for the lack of mathematical procedure for calculating the Li^+ transference number. The Li^+ transference number (t_{Li^+}) was measured on Li/Li-CuMH SSE/Li symmetric cells by the chronoamperometry test. t_{Li^+} can be evaluated using the following equation:^[19]

$$t_{\text{Li}^+} = \frac{I_{\text{ss}}}{I_0} \times \frac{V - I_0 R_0}{V - I_{\text{ss}} R_{\text{ss}}}$$

where V , I_0 , I_{ss} , R_0 , R_{ss} represent the applied voltage (10 mV), the initial and steady-state currents and the impedance before and after polarization, respectively. Notably, the chronoamperometry test was operated between the EIS tests. As demonstrated in **Figure 3g**, the V , I_0 , I_{ss} , R_0 , and R_{ss} of the Li-CuMH SSE are 10 mV, 19.476 μA , 16.276 μA , 267 ohm and 295 ohm, respectively, thus t_{Li^+} of the Li-CuMH SSE can be deduced to 0.77 (**Table S1**).

Table S1. The parameters measured by i - t curves and EIS for calculating the Li^+ transference number. This table refers to **Figs. 3g** and **S28**.

	ΔV (mV)	I_0 (μA)	I_{ss} (μA)	R_0 (ohm)	R_{ss} (ohm)	t_{Li^+}
Li-CuMH	10	19.476	16.276	267	295	0.77
Liquid electrolyte	10	120	93	78	87	0.26
Pure PEO	10	6.578	3.020	1275	1552	0.16

I_0 and I_{ss} are initial and stable current (μA) during polarization. R_0 and R_{ss} are the impedance (ohm) before and after polarization.

New Figure 3g

Figure 3. g) Li^+ transference number measurement of the Li-CuMH SSE.

Q5: It is important to investigate the cause of sudden drops at 1.0 mA cm^{-2} during galvanostatic cycling with step increased current densities to evaluate the durability of Li-CuMH SSE. Could it be attributed to complete destruction of the Li-CuMH SSE? Performing post-cycling experiments after applying the 1.0 mA cm^{-2} current will provide valuable insights.

Our response: Thank you very much for the very professional question. As shown in **Figure 4a**, the sudden drop of voltage at 1.0 mA cm^{-2} indicates the sudden decrease of cell impedance, which is attributed to the short circuit of cell caused by dendritic Li growth at high current density.^[12] In general, the short circuit of solid-state electrolyte based batteries is not attributed to the destruction of the electrolyte itself, especially for Li-CuMH SSE with extremely high chemical and electrochemical stability. According to the review's suggestion, the long-term Li plating/stripping reversibility of the Li/Li-CuMH SSE/Li symmetric cell at 1.0 mA cm^{-2} was separately investigated (**Figure R9**), and the sudden drop of voltage also occurred within 20 h owing to the rapid lithium dendrite growth. Moreover, the post-cycling morphology analysis has been conducted and presented in **Figures S37 and S38**, indicating that the Li metal surface preserves a smooth morphology without obvious cracks and dendrites after cycling in the Li/Li-CuMH SSE/Li cell for 200 h.

Figure 4a

Figure 4. a) Critical current density (CCD) measurement of the Li/Li-CuMH SSE/Li symmetric cell.

Figure R9. Long-term Li plating/stripping reversibility of Li/Li-CuMH SSE/Li symmetric cell at 1 mA cm^{-2} .

Q6: The gradual degradation mechanism of the synthesized solid-state electrolyte in the full cell or in the Li/Li-CuMH SSE/Li configuration should be explained. The authors can investigate the gradual polarization of the Li-CuMH SSE or its swelling by performing *ex situ* EIS measurements after 500 cycles.

Our response: We first thank the reviewer for the very professional suggestion. According to the kind reminder of the reviewer, *ex situ* EIS measurements of Li/Li-CuMH SSE/Li cells and the LFP/Li-CuMH SSE/Li full cells were performed (**Figure R10**). After 500 cycles, the Nyquist plots of both Li/Li-CuMH SSE/Li cell and LFP/Li-CuMH SSE/Li full cell exhibited a slight increase in SEI resistance (R_{SEI}) and charge-transfer resistance (R_{CT}). This clearly demonstrates that the gradual degradation of the cell performance is caused by the accumulation effects of interfacial side reactions.

Figure R10. a) *Ex situ* Nyquist plots of Li/Li-CuMH SSE/Li cells after various cycle numbers at 0.5 mA cm^{-2} . b) *Ex situ* Nyquist plots of the LFP/Li-CuMH SSE/Li full cells at the 0th, 10th, 20th, 30th, 40th, 50th, 60th, 70th, 80th, 90th, 100th, 110th, 120th, 130th, 140th, 150th, and 500th cycle, respectively.

Some technical suggestions:

1. In some instances, the water molecules in hydrated CuMH are referred to as crystallized water, while in other places, they are called structural water. The author needs to use

consistent terminology throughout the manuscript to avoid confusion. It is recommended to clarify and use a single term that accurately reflects the water molecules' coordination to the Cu centers as indicated by the crystallographic evidence.

Our response: We appreciate the reviewer's professional suggestion. As suggested by the reviewer, all expressions of "crystallized water" in our manuscript were revised to "structural water". And the revised texts are marked in red within the revised manuscript.

2. To enhance technical clarity, it would be beneficial to include bond distances between the oxygen atoms from water-Cu and maleate-Cu centers. Additionally, a clear crystallographic figure illustrating the structural building unit of CuMH with all the coordination sites around the Cu center should be provided.

Our response: We first thank the reviewer for the very insightful comment and have provided the related bond distances and coordination environment of the Cu center in the new **Figures S6, S7** and **Table R3** according to the reviewer's suggestion.

"According to the valence information analysis, the crystal structure of Li-CuMH was further calculated and presented in **Fig. S6**. The Cu (II) ion possesses a square pyramidal coordination environment, and the apical oxygen atoms originate from the localized crystallized water molecules. The base atoms come from two oxygen atoms of a single maleate chelating ligand, and two oxygen atoms come from two additional maleate groups. The Cu (I) ion has a similar square pyramidal coordination environment as the Cu (II) ion, with the difference that one of the coordination oxygens is a hydroxyl oxygen, not a conjugated carbonyl oxygen. Each maleate group is bonded to three copper atoms and forms a polymeric monolayer. These polymeric monolayers are closely connected by strong hydrogen bonds between water molecules and the carbonyl oxygen atoms to form a periodic stacked structure (**Fig. S7**)^[38]."

New Figure S6

Figure S6. Coordination structures of Cu^{2+} and Cu^+ with MA in the CuMH crystal, respectively.

New Figure S7

Figure S7. The crystal structure of CuMH containing Cu^+ and Cu^{2+} along the c -axis.

The bond lengths of the Cu-O are in the range from 1.995 to 2.461 Å, indicating the strong coordination of Cu with oxygen atoms in carboxylic acid and structured H_2O .

Table R3. The bond distances between the oxygen atoms from water-Cu and maleate-Cu centers.

Bond distance (Å)	Cu-O (O from carboxylic acid)			Cu-O (O from H_2O)	
Cu (II) in $\text{Cu}^{\text{II}}\text{MH}$	2.058	2.066	2.003	2.261	2.197
Cu (I) in $\text{Cu}^{\text{I}}\text{MH}$	1.995	1.963	2.032	2.461	2.182

3. In the Results and Discussion section, it is advisable to create subheadings such as "CuMH synthesis" and "Li-CuMH SSE film preparation" to better represent the two steps involved in Li-CuMH SSE synthesis.

Our response: We appreciate the reviewer for the rational suggestion. And the relevant parts have been revised on pages 3 and 5 of the revised manuscript according to the reviewer's comment. The new subheadings are "Synthesis of CuMH nanoflakes" and "Fabrication and characterization of Li-CuMH SSEs".

4. The solvent used for the emulsion to prepare the Li-CuMH film with PTFE should be mentioned.

Our response: We appreciate the reviewer for the professional question. The solvent used for the emulsion to prepare the Li-CuMH film with PTFE is water. According to the reviewer's suggestion, the solvent used for polytetrafluoroethylene emulsion is noted on page 2 of the revised supplementary information. We copy the relevant text for your check:

"The flexible and compact CuMH film were fabricated by rolling a viscous paste comprising CuMH powders and polytetrafluoroethylene (PTFE, 60 wt.% in H₂O, Macklin) emulsion in a solid ratio of 9:1."

5. The author should provide the CCDC (Cambridge Crystallographic Data Centre) number for the single crystal structure of CuMH.

Our Response: We thank the reviewer for the very professional suggestion and fully agree that the CCDC number of CuMH should be provided. The CCDC (Cambridge Crystallographic Data Centre) number for the single crystal structure of CuMH is 1133197.^[6] We also provided three CIF files associated with Cu^{II}MH, Cu^{I/II}MH and Li-Cu^{I/II}MH when uploading the revised manuscript.

6. Figure 1a does not clearly explain the synthetic procedure. It would be helpful to include additional information from the text in the figure. The incorporation of LiNO₃ should be explicitly mentioned. The presence of Li should also be identified in this figure.

Our response: We thank the reviewer for the very professional question. As demonstrated in

Figure 1a, a fresh copper (Cu) foil was directly immersed in an acetonitrile (AN) solution containing an appropriate amount of maleic acid (MA) and lithium nitrate (LiNO_3) under atmospheric environment, in which MA not only acts as a ligand, but also releases H^+ to cooperate with NO_3^- to promote the dissolution of copper. Therefore, the role of a small amount of LiNO_3 is to provide NO_3^- ions to promote the dissolution of copper, while Li^+ is relatively unimportant. To emphasize the role of LiNO_3 , the synthesis of CuMH nanoflakes was photographed at different reaction times to study the reaction kinetics with and without LiNO_3 . As shown in the new **Figure S1**, the production speed of CuMH has been greatly accelerated with the aid of LiNO_3 , while the reaction rate without LiNO_3 is extremely low. Note that in our actual experiments, the synthesis speed of CuMH powder under stirring conditions was much faster. As suggested by the reviewer, we have added a short discussion on page 3 in the revised manuscript. We copy the relevant text here for your check:

New Figure S1

Figure S1. Digital photographs of the production speed of CuMH powders at different reaction times a) without and b) with the addition of LiNO_3 . The production speed of CuMH has been greatly accelerated with the aid of LiNO_3 , while the reaction rate without LiNO_3 is extremely low.

“Briefly, a fresh copper (Cu) foil was directly immersed in an acetonitrile (AN) solution containing an appropriate amount of maleic acid (MA) and lithium nitrate (LiNO_3) under atmospheric environment, in which MA not only acts as ligand, but also releases H^+ to cooperate with NO_3^- to promote the dissolution of copper (the significant acceleration effect of NO_3^- on CuMH production is presented in **Fig. S1**).”

7. *It is recommended to replace Figure 1d with a new photograph that clearly shows both the unreacted Cu-foil and the Li-CuMH coated portion in a single picture to provide a more realistic representation.*

Our response: We appreciate the reviewer for the thoughtful advice. In our manuscript, **Figure 1d** provides the digital photograph of a typical 6 cm × 6 cm-sized Li-CuMH SSE film. According to the reviewer's suggestion, we added the new **Figure S1** in the revised supporting information, in which unreacted Cu-foil and the *in situ* generated blue CuMH powders are in a single picture to visually show the synthesis process of the CuMH (please see our answer to Q6).

8. *The claim of increased dissociation content and dissociation effect of LiTFSI in Li-CuMH based on the narrow Raman shift window by deconvolution is an overestimation. The author should avoid making such exaggerated claims.*

Our response: We first thank the reviewer for the very professional suggestion and apologize for the overestimation. According to the reviewer's comments, the relevant part has been corrected as a brief description on page 10 of the revised manuscript. **We copy the relevant text here for your check:**

“In order to identify specific forms of TFSI⁻ coordination to the lithium cations, the expansion and contraction modes of the entire TFSI⁻ anion at 750 cm⁻¹ was further analyzed using Raman spectroscopy (**Fig. 3h**), which could produce large polarization changes^[45]. This wave band can be divided into three vibrational components located at 740, 750 and 756 cm⁻¹, respectively derived from free anion, TFSI⁻ coordination to one Li⁺ cation (CIP) and TFSI⁻ coordination to two or more Li⁺ cations (AGGs)^[46]. Apparently, more free TFSI⁻ anions can be released from Li-CuMH compared with the organic liquid electrolyte containing 1M LiTFSI, indicating an increased dissociation content of LiTFSI salt in Li-CuMH. Interestingly, the AGGs signal completely disappears in the Li-CuMH SSE spectrum, further demonstrating the optimized dissociation effect. FT-IR analysis could also confirm the increase in free TFSI⁻ content in the Li-CuMH SSE (**Fig. S26**). These results provide another supporting evidence that electronegative COO⁻ groups and structural water in Li-CuMH can preferentially coordinate

with Li⁺ ions, promoting the dissociation of LiTFSI salts.”

Figure 3. h) Raman spectra of the TFSI⁻ vibration of 1M LiTFSI in liquid DOL/DME electrolyte and Li-CuMH SSE.

9. In line 411, it is most likely referring to powder not power.

Our response: We appreciate the reviewer for the professional suggestion. We are sorry for the format error. According to the reviewer’s kind reminding, the spelling mistake has been corrected.

10. In line 419, it should be either "success of embedding" or "successful embedding."

Our response: We appreciate the reviewer for the professional suggestion. We are sorry for the format error. According to the reviewer’s kind reminding, the grammatical mistake has been corrected.

11. The statement "Li-CuMH SSE optimizes the interfacial SEI layer" is a valuable inclusion. However, to maintain the readers' focus on the main topic and to prevent unnecessary elongation of the manuscript, it is suggested to move this portion to the supporting information.

Our response: We appreciate the reviewer for the very instructive suggestion. According to the reviewer’s suggestion, we have greatly simplified the **Figure 6** in the main text and transferred a considerable portion of the content to the supporting information. The revised

new Figure 6 is shown below.

New Figure 6

Figure 6. Optimized SEI layer on the Li metal surface. a) 3D AFM images (scan size: $2 \mu m \times 2 \mu m$) and b) the corresponding force-separation curves of Li anodes disassembled from Li/Li symmetric cells with Li-CuMH SSE. c) F 1s, N 1s and C 1s in-depth XPS spectra of SEI layers on Li anodes disassembled from Li/Li symmetric cells with Li-CuMH SSE. d, e) TOF-SIMS 3D reconstruction of LiF_2^- , $C_2H_2O^-$ and CO_3^- species of SEI layers on Li anodes disassembled from Li/Li symmetric cells with Li-CuMH SSE and liquid electrolyte. f, g) Schematics of SEI composition affected by Li-CuMH SSE and liquid electrolyte.

Reviewer #3 (Remarks to the Author): =====

This article reports an interesting work on the development of SPE for solid state batteries. Before accepting, authors should revise the introduction section to include recent works, include more details on sample preparation, compare battery performance with the literature.

Our response: We appreciate the reviewer's very positive and professional comments and fully agree that the most recent literatures should be added and compared. The revisions from the reviewer have been fully considered and thoroughly addressed, as shown below.

According to the reviewer's suggestion, we have introduced the following latest literature in the introduction:

[7] Ma, Y. et al. Scalable, ultrathin, and high-temperature-resistant solid polymer electrolytes for energy-dense lithium metal batteries. *Adv. Energy Mater.* 12, 2103720 (2022).

[8] Tang, L. et al. Polyfluorinated crosslinker-based solid polymer electrolytes for long-cycling 4.5 V lithium metal batteries. *Nat. Commun.* 14, 2301 (2023).

[20] Huo, H. et al. Rational design of hierarchical "ceramic-in-polymer" and "polymer-in-ceramic" electrolytes for dendrite-free solid-state batteries. *Adv. Energy Mater.* 9, 1804004 (2019).

[21] Wen, S. et al. Integrated design of ultrathin crosslinked network polymer electrolytes for flexible and stable all-solid-state lithium batteries. *Energy Storage Mater.* 47, 453-461 (2022).

[22] Yang, K. et al. Stable interface chemistry and multiple ion transport of composite electrolyte contribute to ultra-long cycling solid-state $\text{LiNi}_{0.8}\text{Co}_{0.1}\text{Mn}_{0.1}\text{O}_2$ /lithium metal batteries. *Angew. Chem. Int. Ed.* 60, 24668-24675 (2021).

[23] Xu, S. et al. Homogeneous and fast ion conduction of PEO-based solid-state electrolyte at low temperature. *Adv. Funct. Mater.* 30, 2007172 (2020).

[28] Wu, N. et al. Fast Li^+ conduction mechanism and interfacial chemistry of a nasicon/polymer composite electrolyte. *J. Am. Chem. Soc.* 142, 2497-2505 (2020).

As requested by the reviewer, we have added more synthesis details of the CuMH nanoflakes in the supplementary information. We copy the relevant text here for your check:

“Synthesis of copper maleate monohydrate (CuMH) nanoflakes

Copper maleate monohydrate (CuMH) nanoflakes with controllable 2D morphology were

prepared by the following procedure. Briefly, a fresh Cu foil (99.999% purity) was directly placed in an acetonitrile (AN, Guoyao, GR) solution dissolved in 6 mM maleic acid (MA, TCI, 99.0% purity) and 0.36 mM lithium nitrate (LiNO_3) under atmospheric environment. During standing for 24 hours, the blue precipitate was continuously deposited in the reaction container. After that, the precipitates were collected by centrifugation, washed with AN solvent to remove residual MA and LiNO_3 for three times and vacuum dried at 60 °C for 24 hours prior to use.”

In the new **Figure 4b**, we provide a comprehensive comparison of CCD and overpotential values between the Li-CuMH SSE and reported solid-state electrolytes.

New Figure 4b

Figure 4. b) Comparison of CCD and overpotential values between the Li-CuMH SSE and reported solid-state electrolytes^[29, 47-52].

In **Tables S2** and **S5**, the Li/Li symmetric cells and LFP/Li batteries with the Li-CuMH SSE and other SSEs in the literatures were systematically listed and compared.

Table S2. Li/Li symmetric cell performance of Li-CuMH SSE and SSEs in the literature. This table refers to **Figure 4b**.

Electrolyte	Type	Current density (mA cm^{-2})	Overpotential (mV)	Ref.
-------------	------	---	--------------------	------

Li-Cu-CNF	Cellulose	0.5	~100	[29]
21- β -CD-g-PTFEMA	Topological polymer	0.1 (70 °C)	100	[47]
LIMIC-15	Rigid-rod polymer	0.2	~200	[48]
PEO/Mg (ClO ₄) ₂	PEO-based	0.4 (55 °C)	~300	[49]
PEO/Li ₂ S ₆	PEO-based	0.2 (45 °C)	~300	[50]
HKUST-1	MOF-based	0.125	~20	[51]
CD-COF-Li	COF-based	0.2	40	[52]
Li-CuMH SSE	2D lamellar	0.9	284	This work

Table S5. Comparison of electrochemical performance of LFP/Li batteries with the Li-CuMH SSE and SSEs previously reported in the literatures.

Electrolyte	Type	Current density	Capacity retention	Ref.
Li-Cu-CNF	Cellulose	0.1 C	200 cycles, 94%	[29]
PEO/Li ₂ S ₆	PEO-based	0.1 mA cm ⁻² (50 °C)	700 cycles, 89.2%	[50]
HKUST-1	MOF-PTFE	1 C	500 cycles, 75%	[51]
CD-COF-Li	COF-based	0.1 C	100 cycles, 91%	[52]
LLZO/PEGMEA	Asymmetric layers	0.2 C	120 cycles, 94.5%	S9 ^[9]
LZONs/PEO	Solid-polymer-solid	0.1 C	1500 cycles, 70%	S10 ^[10]
Li-RCC1-ClO ₄	Organic ionic cage	1 C	750 cycles, 88.2%	S11 ^[11]
Li-CuMH SSE	2D lamellar	0.5 C	200 cycles, 90% 573 cycles, 80%	This work

Reference in response letter:

- [1] Liu, P. & Hensen, E.J.M. Highly efficient and robust Au/MgCuCr₂O₄ catalyst for gas-phase oxidation of ethanol to acetaldehyde. *J. Am. Chem. Soc.* **135**, 14032-14035 (2013).
- [2] Deutsch, K.L. & Shanks, B.H. Active species of copper chromite catalyst in C–O hydrogenolysis of 5-methylfurfuryl alcohol. *J. Catal* **285**, 235-241 (2012).
- [3] Severino, F., Brito, J.L., Laine, J., Fierro, J.L.G. & Agudo, A.L. Nature of copper active sites in the carbon monoxide oxidation on CuAl₂O₄ and CuCr₂O₄ spinel type catalysts. *J. Catal* **177**, 82-95 (1998).
- [4] Platzman, I., Brener, R., Haick, H. & Tannenbaum, R. Oxidation of polycrystalline copper thin films at ambient conditions. *J. Phys. Chem. C* **112**, 1101-1108 (2008).
- [5] Chen, R., Li, Q., Yu, X., Chen, L. & Li, H. Approaching practically accessible solid-state batteries: Stability issues related to solid electrolytes and interfaces. *Chem. Rev.* **120**, 6820-6877 (2019).
- [6] Prout, C.K., Carruthers, J.R. & Rossotti, F.J.C. Structure and stability of carboxylate complexes. Part viii. Crystal and molecular structures of copper(ii) hydrogen maleate tetrahydrate and copper(ii) maleate hydrate. *Journal of the Chemical Society A: Inorganic, Physical, Theoretical* 3342-3349 (1971).
- [7] Guo, D. et al. Foldable solid-state batteries enabled by electrolyte mediation in covalent organic frameworks. *Adv. Mater.* **34**, 2201410 (2022).
- [8] Seo, D.-H., Kim, H., Kim, H., Goddard, W.A. & Kang, K. The predicted crystal structure of Li₄C₆O₆, an organic cathode material for Li-ion batteries, from first-principles multi-level computational methods. *Energy Environ. Sci.* **4**, 4938-4941 (2011).
- [9] Banerjee, A., Araujo, R.B., Sjödin, M. & Ahuja, R. Identifying the tuning key of disproportionation redox reaction in terephthalate: A Li-based anode for sustainable organic batteries. *Nano Energy* **47**, 301-308 (2018).
- [10] Viswanathan, L. & Virkar, A.V. Wetting characteristics of sodium on β"-alumina and on nasicon. *J. Mater. Sci.* **17**, 753-759 (1982).
- [11] Ruan, Y. et al. A 3D cross-linking lithiophilic and electronically insulating interfacial engineering for garnet-type solid-state lithium batteries. *Adv. Funct. Mater.* **31**, 2007815 (2020).
- [12] Lu, Y. et al. Critical current density in solid-state lithium metal batteries: Mechanism,

influences, and strategies. *Adv. Funct. Mater.* **31**, 2009925 (2021).

[13] Binitha, M.P. & Pradyumnan, P.P. Spectroscopic, thermal and dielectric studies on copper maleate monohydrate single crystals. *Indian J. Pure Appl. Phys.* **51**, 453-457 (2016).

[14] Yang, C. et al. Copper-coordinated cellulose ion conductors for solid-state batteries. *Nature* **598**, 590-596 (2021).

[15] Hu, W., Peng, Y., Wei, Y. & Yang, Y. Application of electrochemical impedance spectroscopy to degradation and aging research of lithium-ion batteries. *J. Phys. Chem. C* **127**, 4465-4495 (2023).

[16] Liu, J. et al. Cationic covalent organic framework with ultralow homo energy used as scaffolds for 5.2 V solid polycarbonate electrolytes. *Adv. Sci.* **9**, 2200390 (2022).

[17] Niu, C., Luo, W., Dai, C., Yu, C. & Xu, Y. High-voltage-tolerant covalent organic framework electrolyte with holistically oriented channels for solid-state lithium metal batteries with nickel-rich cathodes. *Angew. Chem. Int. Ed.* **60**, 24915-24923 (2021).

[18] Pan, J. et al. A quasi-double-layer solid electrolyte with adjustable interphases enabling high-voltage solid-state batteries. *Adv. Mater.* **34**, 2107183 (2022).

[19] Bruce, P.G. & Vincent, C.A. Steady state current flow in solid binary electrolyte cells. *Journal of Electroanalytical Chemistry and Interfacial Electrochemistry* **225**, 1-17 (1987).

REVIEWER COMMENTS

Reviewer #1 (Remarks to the Author):

The revised manuscript is greatly improved. I was extremely skeptical after reading the first draft that there was any lithium intercalation into these materials, but the authors did an excellent job of digesting my criticisms and identifying informative experiments to perform. The small shifts observed in the XRD patterns with increasing soaking time do indeed support the intercalation of substoichiometric amounts of Li⁺ into the material. The authors should point out that these small changes do suggest substoichiometric amounts of Li, since whole-number Li atoms per unit cell should give a more drastic change in cell parameters. The substoichiometric nature of the Li intercalation also satisfies my concern about massive charge separation that would occur if stoichiometric amounts of Li salts were to dissociate, requiring massive charge separation. I am also grateful to the authors for acknowledging the cause of the original identical spectra: air decomposition, and that they solved the problem by moving to the glove box.

The use of Nyquist plots as a function of soaking time, and discussing these in the context of the changes in XRD patterns is also a creative and compelling addition.

The addition of XPS and Auger spectra to support a mixture of Cu(I) and Cu(II) species (and hence, infer the presence of exchangeable protons) is very resourceful! The exchange of the H⁺ with Li⁺ further satisfies concerns about charge separation, though it would be nice if there was some way to directly support the presence/exchange of H⁺, rather than inferring it from copper spectra.

Thanks also for explaining the utility of the CCD analysis.

My confidence that this work is appropriate for Nature Comm. is increased based on this revision. However there are still a few things I would like to see some clarification on, listed below:

1. I am unclear on how the crystal structure of this material is being determined. In the first draft I (incorrectly?) concluded that it was a single-crystal X-ray structure solution, which was why I asked for CIFs. However, the CIFs the authors provide are VESTA CIFs, which makes it unclear whether these are experimental cifs, or just electronic versions of the structural models. Did the authors solve this structure? Was there a single crystal or was it determined from powder? (I see no details on Rietveld refinement). Was it previously determined by another group? I don't see a citation. How did the authors determine this structure?

2. Is there any way to obtain direct evidence for the presence of exchangeable protons in the precursor? Here are a few ideas:

- Can the authors detect differences in the FT-IR assignable to protonated maleates before and after Li exchange?

- Can the authors try exchanging Li into these materials in water solvent or non-aqueous solvent to see if the pH drops? Is there a pH indicator they could use in non-aqueous solvent to detect the exit of protons from the interlayer?

- The authors used Li NMR to detect changes in Li environment. Could ^1H NMR shifts be used to argue for differences in the proton environments before and after lithium exchange?

3. Refer to XRD patterns as "patterns" instead of "spectra."

If the authors could address 1-3 I believe I would be satisfied and recommend publication.

Reviewer #2 (Remarks to the Author):

Dear Authors,

Thank you for the thorough review. Apart from a few minor errors (such as crystallized water in some places), I am satisfied with the revised manuscript and the detailed responses to all the questions.

Many thanks.

Reviewer #3 (Remarks to the Author):

All comments have been included in the manuscript

Point-by-point Response to Review Comments

We sincerely thank the reviewers for their thoughtful comments and valuable suggestions, which significantly improved this paper. Based on their feedback, we made comprehensive revisions to the manuscript to enhance its comprehensiveness, concision and coherence. The reviewers' suggestions are valuable and instructive for our scientific research. All revisions have been incorporated into the revised manuscript. To clearly address the reviewers' comments, we categorized them into specific question areas and provided detailed responses as follows.

Reviewers' comments and our Response:

Reviewer #1 (Remarks to the Author): =====

The revised manuscript is greatly improved. I was extremely skeptical after reading the first draft that there was any lithium intercalation into these materials, but the authors did an excellent job of digesting my criticisms and identifying informative experiments to perform. The small shifts observed in the XRD patterns with increasing soaking time do indeed support the intercalation of substoichiometric amounts of Li⁺ into the material. The authors should point out that these small changes do suggest substoichiometric amounts of Li, since whole-number Li atoms per unit cell should give a more drastic change in cell parameters. The substoichiometric nature of the Li intercalation also satisfies my concern about massive charge separation that would occur if stoichiometric amounts of Li salts were to dissociate, requiring massive charge separation. I am also grateful to the authors for acknowledging the cause of the original identical spectra: air decomposition, and that they solved the problem by moving to the glove box.

The use of Nyquist plots as a function of soaking time, and discussing these in the context of the changes in XRD patterns is also a creative and compelling addition.

The addition of XPS and Auger spectra to support a mixture of Cu(I) and Cu(II) species (and

hence, infer the presence of exchangeable protons) is very resourceful! The exchange of the H^+ with Li^+ further satisfies concerns about charge separation, though it would be nice if there was some way to directly support the presence/exchange of H^+ , rather than inferring it from copper spectra.

Thanks also for explaining the utility of the CCD analysis.

Our response: We greatly appreciate your recognition of our efforts and valuable feedback on the manuscript. In response to the reviewer's comments, we have meticulously revised the manuscript and provided detailed point-by-point responses as outlined below.

My confidence that this work is appropriate for Nature Comm. is increased based on this revision. However there are still a few things I would like to see some clarification on, listed below:

Q1: *I am unclear on how the crystal structure of this material is being determined. In the first draft I (incorrectly?) concluded that it was a single-crystal X-ray structure solution, which was why I asked for CIFs. However, the CIFs the authors provide are VESTA CIFs, which makes it unclear whether these are experimental cifs, or just electronic versions of the structural models. Did the authors solve this structure? Was there a single crystal or was it determined from powder? (I see no details on Rietveld refinement). Was it previously determined by another group? I don't see a citation. How did the authors determine this structure?*

Response: We thank the reviewer for the very professional question. According to review's comments, we further performed Rietveld refinement analyses of powder XRD patterns and obtained CIFs for $Cu^{I/II}MH$ and $Li-Cu^{I/II}MH$. The previously provided VESTA CIFs were obtained by optimizing the single-crystal structure (CuMH-CCDC.cif, PDF#49-2453) on DFT calculations. The new CuMH.cif and Li-CuMH.cif are provided as refined structures.

As shown in new **Fig. S7** and new **Table S1**, the fitted curves are well consistent with the original XRD patterns, corresponding to R_{wp} of 7% and R_p of 5% for $Cu^{I/II}MH$ and R_{wp} of 8%

and R_p of 6% for Li-Cu^{I/II}MH. As a result, the Rietveld refinement confirmed that the maleic acid in the Cu^{I/II}MH structure contains part of H and the Li-Cu^{I/II}MH structure has about 58.3% of Li-ions (**Fig. S12**). In addition, the crystal structures of Cu^{I/II}MH and Li-Cu^{I/II}MH exhibit low structural symmetry, which can be attributed to the partial occupation of H and Li.

The synthesis and characterization of single-crystal CuMH was conducted as early as 1971 by Rossotti *et al.*,^[1] but for the first time in our work, we synthesized CuMH powders containing partial H in maleic acid. In addition to the powder XRD refinement, liquid ¹H NMR (**Fig. S5**) and FTIR spectroscopy (**Fig. S13**) also convincingly confirmed that some maleic acid molecules coordinated with copper contain H atoms.

New Figure S7

Figure S7. Rietveld refinement of powder XRD patterns for the Cu^{I/II}MH (a) and Li-Cu^{I/II}MH (b) samples. The observed and calculated intensities are shown as the red dots and the black solid line, respectively. The bottom blue line shows the fitting residual difference. The Bragg positions are represented by purple scale lines.

Table S1. Structural analysis results obtained from Rietveld refinement XRD patterns of the Cu^{I/II}MH and Li-Cu^{I/II}MH samples.

Samples	Space group	$a/\text{Å}$	$b/\text{Å}$	$c/\text{Å}$	$V/\text{Å}^3$	$\alpha/^\circ$	$\beta/^\circ$	$\gamma/^\circ$	R_{wp} (%)	R_p (%)
---------	-------------	--------------	--------------	--------------	----------------	-----------------	----------------	-----------------	--------------	-----------

Cu ^{I/II} MH	P1	15.512	15.857	7.713	1750.552	90.735	112.660	89.583	7.45	5.24
Li-Cu ^{I/II} MH	P1	15.574	16.023	7.759	1792.155	90.070	112.240	89.980	8.43	6.50

New Figure S5

Figure S5. a) ¹H NMR spectra of original CuMH sample and the CuMH sample after the hydrogen/deuterium (H/D) exchange. b) ¹H NMR spectrum of the Li-CuMH sample.

Figure S12. The crystal structure of Li-CuMH produced by the ion-exchange reaction between Li⁺ ions and H⁺ from CuMH.

According to the reviewer's professional suggestions, we have revised the relevant parts of page 5 of the revised manuscript, as well as pages 6 and 12 of the supplementary materials.

“Powder X-ray diffraction (PXRD) pattern of the as-obtained CuMH powder is shown in **Fig. S6** and single-crystal XRD pattern of the CuMH grown in water perfectly matches with the standard powder diffraction card (PDF#49-2453). To further determine the crystal structures of Cu^{I/II}MH and Li-Cu^{I/II}MH samples, the General Structure Analysis System (GSAS) program was used to solve and refine the corresponding PXRD patterns by the Rietveld method^[40], which confirms the existence of protons in the CuMH crystal and the Li⁺/H⁺ exchange reaction for the Li-CuMH sample (**Fig. S7** and **Table S1**).”

“Single-crystal X-ray diffraction pattern was collected on XtaLAB Synergy (Dualflex; HyPix) and XtaLAB Synergy R (DW system; HyPix) diffractometers using Cu K α ($\lambda = 1.54184 \text{ \AA}$) micro-focus X-ray sources (PhotonJet (Cu) X-ray source). The CrysAlisPro software was used to collect and reduce the raw data. The structures were solved by ShelXT with intrinsic phasing and refined on F^2 using full-matrix least-squares methods, with ShelXL and Olex2 used as graphical user interfaces.”

New Figure S7

Figure S7. Rietveld refinement of powder XRD patterns for the Cu^{I/II}MH (a) and Li-Cu^{I/II}MH (b) samples. The observed and calculated intensities are shown as the red dots and the black solid line, respectively. The bottom blue line shows the fitting residual difference. The Bragg positions are represented by purple scale lines.

Table S1. Structural analysis results obtained from Rietveld refinement XRD patterns of the

Cu^{I/II}MH and Li-Cu^{I/II}MH samples.

Samples	Space group	a/ Å	b/ Å	c/ Å	V/ Å ³	α /°	β /°	γ /°	R_{wp}	R_p
									(%)	(%)
Cu ^{I/II} MH	P1	15.512	15.857	7.713	1750.552	90.735	112.660	89.583	7.45	5.24
Li-Cu ^{I/II} MH	P1	15.574	16.023	7.759	1792.155	90.070	112.240	89.980	8.43	6.50

Q2: Is there any way to obtain direct evidence for the presence of exchangeable protons in the precursor? Here are a few ideas:

- Can the authors detect differences in the FT-IR assignable to protonated maleates before and after Li exchange?

Our response: We thank the reviewer for the very rigorous question and fully agree with the reviewer's point. The chemical bonding information of the CuMH and Li-CuMH samples was identified by the FT-IR spectra. As manifested in **Fig. S13**, the absorption peaks located at 1540, 1580/1433, 682/620 cm⁻¹ can be identified to -COOH group, COO⁻ group and Cu-O bond of the CuMH sample, respectively.^[2, 3] Note that the characteristic peak of -COOH group completely disappears in the FT-IR spectrum of the Li-CuMH sample, and the new peak at 870 cm⁻¹ is assigned to the Li-O bond.^[4] Therefore, the FT-IR spectra confirm the coexistence of -COOH and COO⁻ groups in the CuMH lattice.

New Figure S13

Figure S13. FT-IR spectra of CuMH and Li-CuMH samples.

According to the reviewer's insightful comments, we have revised the relevant parts on page 6 of the revised manuscript, and on page 15 of the supplementary information.

“Fourier transform infrared (FT-IR) measurements were performed to analyze the chemical environment in the CuMH and Li-CuMH crystals, as well as the existing form of structural water molecules. As illustrated in **Fig. S13**, the FT-IR peaks located at 1540, 1580/1433, 682/620 cm^{-1} can be identified to -COOH group, COO^- group and Cu-O bond of the CuMH sample, respectively^[44, 45]. Note that the characteristic peak of -COOH group completely disappears in the FT-IR spectrum of the Li-CuMH sample and the new peak at 870 cm^{-1} is assigned to the Li-O bond^[46]. The broad band in the wavenumber range of 3200 to 3500 cm^{-1} can be identified as the O-H stretching vibration of structural water, and the band at around 1654 cm^{-1} corresponds to the H-O-H bending mode in the lattice (**Fig. S13**). The FT-IR spectra confirmed the coexistence of -COOH, COO^- groups and structural H_2O in the CuMH lattice, which is highly consistent with what we found in ^1H NMR analysis (**Fig. S5**).”

Figure S13. FT-IR spectra of CuMH and Li-CuMH samples.

- Can the authors try exchanging Li into these materials in water solvent or non-aqueous solvent to see if the pH drops? Is there a pH indicator they could use in non-aqueous solvent to detect the exit of protons from the interlayer?

Our Response: We greatly appreciate the reviewer for the insightful comment. The pH meter is used to measure the concentration of the hydronium ions (H_3O^+) in the aqueous solutions. However, the CuMH sample is hardly soluble in the H_2O solvent (**Figure R1**). Note that **Figure RX** represents the figures used in the response letter and will not appear in the manuscript and supporting information. Hence, it's difficult to detect the existence of protons through pH measurement. Fortunately, the CuMH sample is soluble in the dimethyl sulfoxide (DMSO) solvent. Therefore, the liquid ^1H nuclear magnetic resonance (NMR) can be used to detect the existence of protons from the interlayer in the CuMH crystal in this work.

Figure R1. Digital photograph of the CuMH powder in the H_2O solvent.

- The authors used Li NMR to detect changes in Li environment. Could ^1H NMR shifts be used to argue for differences in the proton environments before and after lithium exchange?

Response: We greatly appreciate the reviewer for the very professional and insightful comments. Notably, the CuMH sample can be dissolved well in the DMSO solvent (**Figure R2**). Hence, the proton environments of the CuMH and Li-CuMH samples were measured by liquid ^1H nuclear magnetic resonance (NMR), and the deuterated DMSO (DMSO-d_6) was used as the solvent. As shown in new **Fig. S5a**, ^1H NMR signals of the original CuMH sample present two characteristic peaks at ~ 12.1 and 6.2 ppm, corresponding to resonances of hydroxyl hydrogens ($-\text{COOH}$)^[5] and the methylene protons ($-\text{CH}=\text{CH}-$)^[6], respectively. The ^1H NMR signal of the hydroxyl hydrogen ($-\text{COOH}$) was further confirmed by the hydrogen/deuterium (H/D) exchange experiment (**Equation R1**).

The ^1H NMR spectrum of the CuMH sample with addition of 1-2 drop D_2O exhibits two peaks at 6.2 and 5.5 ppm, which are assigned to resonances of the methylene protons ($-\text{CH}=\text{CH}-$) and protons of HOD, while the peak at 12.1 ppm completely disappeared after H/D exchange, convincingly indicating the existence of hydroxyl hydrogens ($-\text{COOH}$) in the CuMH sample. Moreover, the ^1H NMR spectrum of the Li-CuMH sample only shows one characteristic peak of the methylene protons ($-\text{CH}=\text{CH}-$) at 6.2 ppm, since the H^+ ions on the carboxylic acid of CuMH underwent an ion-exchange reaction with the Li^+ ions during the soaking process (new **Fig. S5b**).

Figure R2. Digital photograph of CuMH dissolved in the DMSO solvent.

New Figure S5

Figure S5. a) ^1H NMR spectra of original CuMH sample and the CuMH sample after the hydrogen/deuterium (H/D) exchange. b) ^1H NMR spectrum of the Li-CuMH sample.

According to the reviewer's constructive suggestions, we have added the relevant parts on page 5 of the revised manuscript, and on pages 7, 10 and 11 of the supplementary information:

“The appearance of Cu^+ ions indicates that some undissociated H^+ ions exist on the carboxylic acid that coordinates with Cu^+ ions, resulting in the presence of $(\text{HOCC}_2\text{H}_2\text{COO})^-$ in the structure of the CuMH crystal, which can be confirmed by liquid ^1H nuclear magnetic resonance (^1H NMR) measurements (**Fig. S5**).”

“The structures of CuMH and Li-CuMH samples were characterized by nuclear magnetic resonance (NMR, AVANCE NEO 500M Hz, BRUKER). Specifically, deuterated dimethyl sulfoxide (DMSO-d_6 , 99.9%, Energy Chemical) was used as the solvent for the ^1H NMR measurements.”

New Figure S5

Figure S5. a) ^1H NMR spectra of original CuMH sample and the CuMH sample after the hydrogen/deuterium (H/D) exchange. b) ^1H NMR spectrum of the Li-CuMH sample.

“As shown in **Fig. S5a**, the characteristic peaks in the ^1H NMR spectrum of the pristine CuMH sample are assigned to resonances of hydroxyl hydrogens ($-\text{COOH}$)^[9] and the methylene protons ($-\text{CH}=\text{CH}-$)^[10], respectively. After the hydrogen/deuterium (H/D) exchange reaction, the peaks appearing at 6.2 and 5.5 ppm are identified to resonances of the methylene protons ($-\text{CH}=\text{CH}-$) and protons of HOD, while the peak at 12.1 ppm completely disappears, convincingly demonstrating the existence of hydroxyl hydrogen ($-\text{COOH}$) in the CuMH

sample. Moreover, the ^1H NMR spectrum of the Li-CuMH sample only shows one characteristic peak of the methylene protons (-CH=CH-) at 6.2 ppm, since the H^+ ions on the carboxylic acid of CuMH underwent an ion-exchange reaction with the Li^+ ions during the soaking process (**Fig. S5b**).”

Q3: Refer to XRD patterns as "patterns" instead of "spectra."

Response: We appreciate the reviewer for the professional suggestion. We are sorry for the format errors. According to the reviewer’s kind reminding, we have corrected all expression errors.

If the authors could address 1-3 I believe I would be satisfied and recommend publication.

Reviewer #2 (Remarks to the Author): =====

Dear Authors,

Thank you for the thorough review. Apart from a few minor errors (such as crystallized water in some places), I am satisfied with the revised manuscript and the detailed responses to all the questions.

Many thanks.

Response: We appreciate the reviewer for the professional comments and insightful suggestions. According to the reviewer's kind reminding, we have corrected all expression errors in the revised manuscript.

Reviewer #3 (Remarks to the Author): =====

All comments have been included in the manuscript

Our response: We appreciate the reviewer for the positive recommendation.

Reference in response letter:

- [1] Prout, C.K., Carruthers, J.R. & Rossotti, F.J.C. Structure and stability of carboxylate complexes. Part viii. Crystal and molecular structures of copper(ii) hydrogen maleate tetrahydrate and copper(ii) maleate hydrate. *Journal of the Chemical Society A: Inorganic, Physical, Theoretical* 3342-3349 (1971).
- [2] Mouaïne, K., Becker, P. & Carabatos-Nédelec, C. Thermal and spectroscopic study of dehydration of lithium formate monohydrate single-crystals. *J. Therm. Anal. Calorim.* **55**, 807-816 (1999).
- [3] Hu, Z. et al. Secondary bonding channel design induces intercalation pseudocapacitance toward ultrahigh-capacity and high-rate organic electrodes. *Adv. Mater.* **33**, 2104039 (2021).
- [4] Hu, Z. et al. Self-assembled binary organic granules with multiple lithium uptake mechanisms toward high-energy flexible lithium-ion hybrid supercapacitors. *Adv. Energy Mater.* **8**, 1802273 (2018).
- [5] Malek, K., Vala, M., Kozłowski, H. & Proniewicz, L.M. Experimental and theoretical NMR study of selected oxocarboxylic acid oximes. *Magn. Reson. Chem.* **42**, 23-29 (2003).
- [6] Salem, A.A., Mossa, H.A. & Barsoum, B.N. Quantitative determinations of levofloxacin and rifampicin in pharmaceutical and urine samples using nuclear magnetic resonance spectroscopy. *Spectrochim. Acta. A Mol. Biomol. Spectrosc.* **62**, 466-472 (2005).

REVIEWERS' COMMENTS

Reviewer #1 (Remarks to the Author):

The authors have done a good job of considering and including my comments. This paper is suitable for publication.

Point-by-point Response to Review Comments

Reviewers' comments and our Response:

Reviewer #1 (Remarks to the Author): =====

The authors have done a good job of considering and including my comments. This paper is suitable for publication.

Our response: We are deeply grateful to the reviewer's positive comments. According to the constructive feedback, our research has become more rigorous, refined, and scientifically sound.